# Robust Model Reasoning and Fitting via Dual Sparsity Pursuit

**Xingyu Jiang**[1,2]     **Jiayi Ma**[2*]
[1]Huazhong University of Science and Technology,   [2]Wuhan University
{jiangxy998,jyma2010}@gmail.com

## Abstract

In this paper, we contribute to solving a threefold problem: outlier rejection, true model reasoning and parameter estimation with a unified optimization modeling. To this end, we first pose this task as a sparse subspace recovering problem, to search a maximum of independent bases under an over-embedded data space. Then we convert the objective into a continuous optimization paradigm that estimates sparse solutions for both bases and errors. Wherein a fast and robust solver is proposed to accurately estimate the sparse subspace parameters and error entries, which is implemented by a proximal approximation method under the alternating optimization framework with the "optimal" sub-gradient descent. Extensive experiments regarding known and unknown model fitting on synthetic and challenging real datasets have demonstrated the superiority of our method against the state-of-the-art. We also apply our method to multi-class multi-model fitting and loop closure detection, and achieve promising results both in accuracy and efficiency. *Code is released at*: https://github.com/StaRainJ/DSP.

## 1   Introduction

Geometric model estimation is a fundamental problem in computer vision, serving as the core of many high-level tasks including structure-from-motion [28, 39], SLAM [33, 9], and data alignment [24, 27]. The models to specific data can be explained as line/ellipse/circle for 2D point sets, plane/cylinder/sphere for 3D point sets, or fundamental/homography/affine model for point correspondences extracted from two-view images, *etc.* Estimating the parameters of a predefined model from inliers only is well studied [21]. But real-world inputs unavoidably contain severe noises and outliers, posing great challenges for accurate model estimation. Besides, another key problem rarely considered is *how to recovery the model parameters without knowing the model type*.

Recent researches mainly focus on proposing robust estimators to tackle the impact of noise and outliers, such as regarding it as a Consensus Maximization (CM) problem. This problem can be well solved with **Sample Consensus** (SAC) methods, such as RANSAC [19] and its variants[11, 38, 3, 6, 22]. They commonly sample a smallest inlier set, to best fit a given geometry model following a hypothesize-and-verify strategy [28]. Inside this loop, the model is actually estimated by Direct Linear Transform (DLT) method with a Least-Square (LS) solution. SAC methods can provide probabilistic guarantee of hitting an all-inlier subset. However, this scheme succeeds only if given a predefined model type with sufficient time budget [52]. Another popular strategy is to pose it in a **Global Optimization** framework, which formulates the fitting task as a subspace learning problem [45, 37, 13, 17, 18, 14]. Specifically, the representative method, Dual Principal Component Pursuit (DPCP) [45], tries to minimize an $\ell_1$ co-sparse objective on the sphere. This is further applied to estimate specific models [12, 18], which typically embeds the model as a single hyperplane, or the intersection of multiple hyperplanes, and optimizes globally. This type of methods are admitted with

---

*Corresponding author

37th Conference on Neural Information Processing Systems (NeurIPS 2023), New Orleans, USA.

efficient implementations and strong theoretical guarantees, thus arising great research interest in recent years. **Deep Learning** has also stimulated numerous methods for geometric model learning, which extract deep geometric cues from sparse points using multilayer perceptrons [49, 43, 50], convolutional [51] or graph networks [40]. They can fast output potential inliers once trained, but still require robust estimator such as RANSAC as postprocess for accurate model estimation.

The above mentioned methods can only work on a fact that: *one is certain about the true geometric models, then uses them to guide the formulation for parameter estimation*. But in fact, we can always fit a model less constrained to obtain higher inlier count [31, 37]. Specifically, for homography data, estimating a fundamental matrix may return more consensus points, but many are outliers. While homography estimation can only find the largest plane structure in 3D scene, missing considerable inliers for full motion data. Fitting an unknown model for heavily contaminated data is much challenging and is in general NP hard. Existing methods solve it with model selection criteria [1, 42, 44], which first fit all possible models, then select the best one with a geometric verification, such as widely used GRIC metric [44, 32]. This is what is done in SfM or SLAM pipeline for fundamental matrix and homography identification [39]. However, such strategy is limited by the greedy selection strategy thus requiring huge computational burden. In addition, the used insufficient information would easily cause wrong selection of the true model for constrained motions of camera [37, 31, 32].

In this paper, we will simultaneously solve *i) outlier rejection, ii) true model reasoning* and *iii) parameter estimation* in a unified optimization modeling. To this end, we start with introducing a common paradigm for exact model fitting, then derive our sparse subspace learning theory, which makes it possible to estimate the true model without knowing the model type, and robust to outliers. On this basis, we convert the objective into a continuous optimization paradigm that estimates sparse solutions for both bases and outlier entries. Wherein a fast and robust solution is proposed, that is based on superiorities of projected sub-gradient method (PSGM) and alternating optimization strategy. Finally, the true model and the inliers are directly extracted from our solutions. **Contributions:** i) We are the first to propose a continuous optimization modeling for geometric model fitting with unknown model type and dominant outliers. ii) We propose sparse subspace recovery theory, which is a novel formulation for model reasoning and fitting. iii) We integrate the proximal approximation strategy and sub-gradient descent method into the alternating optimization paradigm, which solves our dual sparsity problem with a convergence rate of $\mathcal{O}(1/k^2)$. iv) Extensive experiments on known/unknown model fitting and two visual applications are designed to validate the superiority of our method.

## 2    Methodology

This paper aims to explore a valid solution for geometric model reasoning and robust fitting. Before this, we first give a brief review of the widely-used solution for exact model estimation, which helps to derive our concerned unknown model fitting problem.

### 2.1    Geometry Preliminaries and New Insights

In the community of model fitting, the objective with geometric error is extremely hard to optimize due to the highly non-linear nature. In contrast, if the data are properly normalized, using linearized error (*i.e.,* algebraic error) to construct objective would show great efficiency to find the optimal solution [18]. For algebraic objective, DLT is known as an efficient method. To be specific, suppose we are given a set of feature correspondences $\mathcal{S} = \{\mathbf{s}_i = (\mathbf{p}_i, \mathbf{p}'_i)\}_{i=1}^N$, where $\mathbf{p}_i = (u_i, v_i, 1)^\top$ and $\mathbf{p}'_i = (u'_i, v'_i, 1)^\top$ are column vectors denoting the homogeneous coordinates of feature points extracted from two-view images. Our goal is to recover the underlying geometric structure including *Fundamental matrix*, *Homography* and *Affine*. This two-view problem is essential in 3D vision applications, and is the main focus of this paper.

**Fundamental Matrix** $\mathbf{F} \in \mathbb{R}^{3 \times 3}$ describes the entire epipolar geometry $\mathbf{p}'^\top_i \mathbf{F} \mathbf{p}_i = 0$. It is suggested to be represented by single equation living in the polynomial space

$$\Phi_{\mathbf{F}}(\mathbf{p}_i, \mathbf{p}'_i)^\top vec(\mathbf{F}) = 0, \tag{1}$$

where

$$\Phi_{\mathbf{F}}(\mathbf{p}_i, \mathbf{p}'_i)^\top = (u'_i u_i, u'_i v_i, u'_i, v'_i u_i, v'_i v_i, v'_i, u_i, v_i, 1), \tag{2}$$

is the embedding of correspondence $(\mathbf{p}_i, \mathbf{p}'_i)$ under epipolar constraint, and $vec(\mathbf{F}) \in \mathbb{R}^9$ is the vector form of matrix $\mathbf{F}$ in row-major order. $N \geq 8$ correspondences can uniquely determine $\mathbf{F}$ up to scale.

**Homography** $\mathbf{H} \in \mathbb{R}^{3 \times 3}$ describes the pure rotation or plane projection, which claims that $\mathbf{p}'_i$ and $\mathbf{H}\mathbf{p}_i$ are co-linear, implying $[\mathbf{p}'_i]_\times \mathbf{H}\mathbf{p}_i = \mathbf{0}$, where $[\mathbf{p}'_i]_\times$ is a skew-symmetric matrix. In DLT solution, it usually converts to the following constraint:

$$\Phi_{\mathbf{H}}(\mathbf{p}_i, \mathbf{p}'_i)^\top vec(\mathbf{H}) = \mathbf{0}, \tag{3}$$

where $vec(\mathbf{H}) \in \mathbb{R}^9$, similarly. And $\Phi_{\mathbf{H}}(\mathbf{p}_i, \mathbf{p}'_i)$ denotes the homography embedding:

$$\Phi_{\mathbf{H}}(\mathbf{p}_i, \mathbf{p}'_i)^\top = \begin{bmatrix} u_i & v_i & 1 & 0 & 0 & 0 & -u'_i u_i & -u'_i v_i & -u'_i \\ 0 & 0 & 0 & u_i & v_i & 1 & -v'_i u_i & -v'_i v_i & -v'_i \end{bmatrix}. \tag{4}$$

This constraint suggests that, $\mathbf{H}$ can be estimated from given at least 4 correspondences.

**Affine Matrix** $\mathbf{A} \in \mathbb{R}^{3 \times 3}$ is the degraded case of $\mathbf{H}$, with the last row of $\mathbf{A}$ being $[0, 0, 1]$. This model implies a linear transformation $\mathbf{p}'_i = \mathbf{A}\mathbf{p}_i$. The affine constraint is also represented as:

$$\Phi_{\mathbf{A}}(\mathbf{p}_i, \mathbf{p}'_i)^\top vec(\mathbf{A}) = \mathbf{0}, \tag{5}$$

where $vec(\mathbf{A}) = [a_{11}, a_{12}, a_{13}, a_{21}, a_{22}, a_{23}, 1]^\top$, and

$$\Phi_{\mathbf{A}}(\mathbf{p}_i, \mathbf{p}'_i)^\top = \begin{bmatrix} u_i & v_i & 1 & 0 & 0 & 0 & -u'_i \\ 0 & 0 & 0 & u_i & v_i & 1 & -v'_i \end{bmatrix} \tag{6}$$

is the affine embedding. Using Eq. (5), the solution can be given by at least 3 correspondences.

Given sufficient inliers, DLT method provides an efficient solution for geometric model fitting, which converts to solving the following minimal least-square problem:

$$\min_\theta \|\mathbf{M}^\top \theta\|_2^2, \quad s.t. \quad \|\theta\|_2 = 1, \tag{7}$$

where $\mathbf{M}$ is the data embedding matrix for specific model $\mathcal{M}$, *i.e.*, $\mathbf{m}_i = \Phi_{\mathcal{M}}(\mathbf{p}_i, \mathbf{p}'_i)$ with $\mathcal{M} \in \{\mathbf{F}, \mathbf{H}, \mathbf{A}\}$, as defined in Eqs. (2), (4) and (6). $\theta = vec(\mathcal{M}) \in \mathbb{R}^D$ indicates the parameter vector of $\mathcal{M}$, and $\|\theta\|_2 = 1$ restricts it in a sphere thus avoiding the trivial solution. The optima solution $\theta^*$ is exactly the right singular vector corresponding to the smallest singular value of $\mathbf{M}$. However, DLT only works on the outlier-free case, due to the use of $\ell_2$ norm. Thus, it is usually applied in SAC paradigm by sampling a clean subset to fit a given model. Another practical formulation is using latent variable inside (7) to indicate the inlier, or using truncated or $\ell_1$ loss [18]. In contrast, the $\ell_1$ objective is easier to optimize due to the convex nature by solving

$$\min_\theta \|\mathbf{M}^\top \theta\|_1, \quad s.t. \quad \|\theta\|_2 = 1. \tag{8}$$

This is actually a subspace learning problem as introduced in DPCP series [45, 52, 13], which can be efficiently solved by a projected sub-gradient method (PSGM) [13]. Besides, [18] gives comprehensive comparisons of above mentioned losses under different model cases. These works also demonstrate the robust property of (8), which roughly states that, the estimation task can even tolerate $O(m^2)$ outliers ($m$ is the inlier number).

**New Insights.** Above formulations almost work on knowing the exact information of model type, such that the embedding matrix $\mathbf{M}$ is correct, *i.e*, choosing $\mathcal{M}$ from $\{\mathbf{F}, \mathbf{H}, \mathbf{A}\}$ to generate $\Phi_{\mathcal{M}}$. Practically, predefining a correct model is much difficult. First, the camera type and motion, or the scene in shooting is not always known. Besides, we cannot manually define the correct model for each image pair in large-scale vision tasks. In this regard, this paper tends to simultaneously seek the true model and robustly estimate the model parameters, by proposing a general continuous optimization framework with dual sparsity constraints.

## 2.2 Dual Sparsity Formulation

In this paper, we aim to estimate the model parameters without knowing the model type. That means we should find a common data embedding $\widetilde{\Phi}(\cdot)$ thus avoiding exact $\Phi_{\mathcal{M}}(\cdot)$ for each model. Instead it converts to constructing model embedding $\Psi(\mathcal{M})$, to generate a common constraint as:

$$\widetilde{\Phi}(\mathbf{p}_i, \mathbf{p}'_i)^\top \Psi(\mathcal{M}) = \mathbf{0}, \quad \mathcal{M} \in \{\mathbf{F}, \mathbf{H}, \mathbf{A}\}. \tag{9}$$

Eq. (9) can provide great advantages for our model reasoning and fitting task. First, it avoids the requirement for exact model type to construct $\Phi_{\mathcal{M}}(\cdot)$ before estimation. Moreover, the solution would have the form of $\Psi(\mathcal{M})$, which directly guides the identification of different models. Eq. (9) essentially relies on the following *Sparse Subspace Recovery* (SSR) theory.

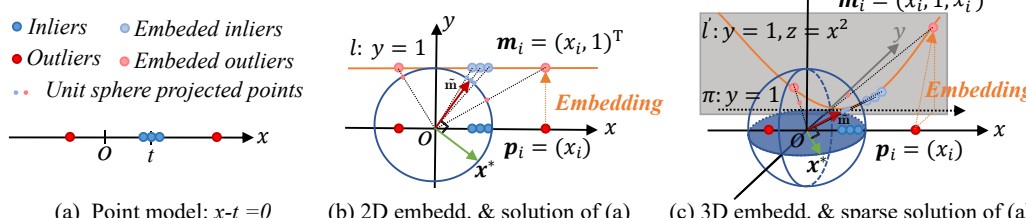

(a) Point model: $x$-$t$ =0      (b) 2D embedd. & solution of (a)      (c) 3D embedd. & sparse solution of (a)

Figure 1: Illustration of fitting a point model ($x = t$) . (a) original points, (b) embedding under DLT solution, (c) embedding under our SSR theory. $\mathbf{x}^*$ or $-\mathbf{x}^*$ is the optimal solution.

**Theorem 1** *(SSR) Geometric model fitting can be seen as a subspace recovery problem represented by the intersection of multiple sparse hyperplanes under an over embedded data space.*

To better interpret the proposed *Theorem 1*, we first give an example on **2D point set fitting**. Given a set of clean 2D points $\mathcal{S} = \{\mathbf{s}_i = (x_i, y_i)\}_{i=1}^{N}$ that are sampled from a structure of *line (L), parabola (P)* or *ellipse (E)*, *i.e.*, $\mathcal{M} \in \{L, P, E\}$. The model parameters can be estimated by exact data embedding $\Phi_{\mathcal{M}}(x_i, y_i)$ with DLT method (7). For ellipse estimation, $\Phi_E(x_i, y_i) = [1, x_i, y_i, x_i^2, y_i^2]^{\top}$, while for a line, $\Phi_L(x_i, y_i) = [1, x_i, y_i]^{\top}$. But in our sparse theory, with a higher-order embedding, such as fixing $\widetilde{\Phi}(x_i, y_i) = \Phi_E(x_i, y_i)$, we can obtain a sparse solution $\theta = [a, b, c, 0, 0]^{\top}$ for a line model, and $\theta = [a, b, c, d, 0]^{\top}$ or $[a, b, c, 0, e]^{\top}$ for a parabola model. See Fig. 2, or refer to *Append.D and Fig. 6* for details. Thus, the SSR task can be formulated as:

$$\min_{\mathbf{x}} \ \|\mathbf{x}\|_0, \qquad s.t. \ \mathbf{M}^{\top}\mathbf{x} = \mathbf{0}, \quad \|\mathbf{x}\|_2 = 1, \tag{10}$$

where $\mathbf{M}$ is the over embedding for input data with $\mathbf{m}_i = \widetilde{\Phi}(x_i, y_i) = \Phi_E(x_i, y_i)$, and we denote $\mathbf{x}$ as a sparse basis that indicates a hyperplane under SSR theory. The $\ell_0$ norm in the objective suggests to use less parameters to fit the given data. Note that, $\mathbf{m}_i$ can be composed by more polynomials such as $x_i y_i$, $x_i^3$, $y_i^3$ or higher-order forms, but it is not necessary for the model pool $\{L, P, E\}$. Problem (10) can be illustrated by a point model fitting $x = t$ as in Fig. 1. The middle plot shows that, the solution $\mathbf{x}^*$ or $-\mathbf{x}^*$ is ideally vertical to the embedding vector $\tilde{\mathbf{m}}_i$ of inliers. With higher-order embedding $[x_i, 1, x_i^2]^{\top}$, the solution would be any vector located in the plane $\mathcal{P}$ that is vertical to $\tilde{\mathbf{m}}_i$ and through the origin. However, with sparsity constraint, the solution reduces to be the intersection of plane $\mathcal{P}$ and one of coordinate planes $\{\pi_{xoy}, \pi_{xoz}, \pi_{yoz}\}$. Since $x_i^2$ causes higher complexity, the final solution is exactly $\mathcal{P} \bigcap \pi_{xoy}$, *i.e.*, $\mathbf{x}^* = \frac{\pm 1}{\sqrt{1+t^2}}[1, -t, 0]^{\top}$.

Denoting $G_{\mathcal{M}} = (D, d, r, s)$ as the geometry relationship for a model $\mathcal{M}$, where $D$ is the ambient dimension of embedded data, $d$ is the subspace dimension, $r$ denotes the number of basis to be estimated, while $s$ means the minimum sparsity in each basis (*i.e.*, the least number of zero parameters), the relations for 2D models are $G_L = (5, 4, 1, 2)$, $G_P = (5, 4, 1, 1)$, and $G_E = (5, 4, 1, 0)$. Next, we emphatically introduce the new formulation for two-view models, which is outlined in *Append.D*.

Extending to **two-view geometry**, our SSR task can directly benefit from (8), which is a special case in subspace recovery [46, 52, 14], with $d = D - 1$. In subspace learning theory, the learning problem is significantly easier when the relative dimension $d/D$ is small [17, 18]. For such purpose, and together with our sparse theory, we can obtain the following new formulations. First, for $\mathbf{F}$ estimation, since its geometric constraint derives only one single hyperplane living in a 9-dimensional space, thus Eq. (9) has the same form with Eq. (1). And the geometric relation is clearly $G_{\mathbf{F}} = (9, 8, 1, 0)$.

**Proposition 1** *Given $n \geq 4$ clean correspondences conforming to a homography transformation $\mathbf{H}$, the model estimation can be converted to recovering a subspace with dimension no more than 6, which consists of 3 sparse hyperplanes with sparsity $s \geq 3$ under the 9-D embedding $\widetilde{\Phi} = \Phi_F(\mathbf{p}_i, \mathbf{p}_i')$.*

*Proof.* Homography model actually derives $r = 3$ equations, and each of them indicates a sparse hyperplane with $\widetilde{\Phi}(\mathbf{p}_i, \mathbf{p}_i')^{\top}\Psi(\mathbf{H}) = \mathbf{0}$, where $\Psi(\mathbf{H}) \in \mathbb{R}^{9 \times 3}$ is the embedding of model $\mathbf{H}$, which consists of three sparse intersected hyperplanes (*i.e.*, orthometric bases) as:

$$\Psi(\mathbf{H}) = \begin{bmatrix} \mathbf{0}_{1 \times 3}, \mathbf{h}_3^{\top}, -\mathbf{h}_2^{\top} \\ -\mathbf{h}_3^{\top}, \mathbf{0}_{1 \times 3}, \mathbf{h}_1^{\top} \\ \mathbf{h}_2^{\top}, -\mathbf{h}_1^{\top}, \mathbf{0}_{1 \times 3} \end{bmatrix}^{\top}, \tag{11}$$

with $\mathbf{H} = [\mathbf{h}_1^\top; \mathbf{h}_2^\top; \mathbf{h}_3^\top]$. Since $\Psi(\mathbf{H})$ is of rank 3, the dimension of subspace $d \leq 6$. And the solution for each basis would have at least $s = 3$ zeros, thus the geometric relation is $G_\mathbf{H} = (9, 6, 3, 3)$. $\quad\square$

**Proposition 2** *Given $n \geq 3$ clean correspondences conforming to an affine transformation $\mathbf{A}$, the model estimation can be converted to recovering a subspace with dimension no more than 7, which consists of 2 sparse hyperplanes with sparsity $s \geq 5$ under the 9-D embedding $\widetilde{\Phi} = \Phi_F(\mathbf{p}_i, \mathbf{p}_i')$.*

*Proof.* Eq. (5) derives $r = 2$ linear equations, and each of them indicates a sparse hyperplane with $\widetilde{\Phi}(\mathbf{p}_i, \mathbf{p}_i')^\top \Psi(\mathbf{A}) = \mathbf{0}$, where $\Psi(\mathbf{A}) \in \mathbb{R}^{9 \times 2}$ is the embedding of model $\mathbf{A}$, which consists of two sparse intersected hyperplanes as:

$$\Psi(\mathbf{A}) = \mathbf{A}_{9\times 2}' = \begin{bmatrix} 0, 0, a_{11}, 0, 0, a_{12}, -1, 0, a_{13} \\ 0, 0, a_{21}, 0, 0, a_{22}, 0, -1, a_{23} \end{bmatrix}^\top . \tag{12}$$

Because $\Psi(\mathbf{A})$ is of rank 2, the dimension of subspace is no more than 7. And the solution for each basis would have at least $s = 5$ zeros, thus the geometric relation is $G_\mathbf{A} = (9, 7, 2, 5)$. $\quad\square$

The above two propositions have well interpreted *Theorem* 1. To be specific, $\mathbf{H}$ and $\mathbf{A}$ estimation can be performed as a hyperplane fitting problem with Eq. (3) and Eq. (5). Alternatively, they can be respectively solved as recovering a 6-dimensional subspace and a 7-dimensional subspace in a common 9-dimensional data embedding space. For the purpose of our unknown model fitting, the advantages of this conversion have been explained in Eq. (9). Besides, the low-relative dimension $d/D$ significantly makes the subspace learning task easier [17, 18]. In the following, we will give a general formulation for unknown model fitting, then explore its efficient solution.

**Formulation for Unknown Model Fitting.** With no outliers and noise, unknown model fitting can convert to finding all $r$ independent sparse bases $\mathbf{X} = [\mathbf{x}_1, \mathbf{x}_2, \cdots, \mathbf{x}_r] \in \mathbb{R}^{D \times r}$ by solving

$$\min_{\mathbf{X} \in \mathbb{R}^{D \times r}} \|\mathbf{X}\|_0, \qquad s.t. \quad \mathbf{M}^\top \mathbf{X} = \mathbf{0}, \quad \text{rank}(\mathbf{X}) = r, \tag{13}$$

where $\mathbf{M}$ is the over embedding matrix of data with $\mathbf{m}_i = \widetilde{\Phi}(\mathbf{s}_i)$. Rank $r$ constraint of $\mathbf{X}$ asks for $r$ independent bases. But $r$ is generally unknown in advance, thus its estimation is also necessary.

**Proposition 3** *Given sufficient inliers with no noise, if the conformed model derives $r$ independent bases, then the solution of SSR satisfies $\max(\text{rank}(\mathbf{X})) = r$.*

*Proof.* Denoting $\mathbf{X}^* = [\mathbf{x}_1, \mathbf{x}_2, \cdots, \mathbf{x}_r] \in \mathbb{R}^{D \times r}$ the optimal solution, any subset of $\mathbf{X}^*$ is also a feasible solution with rank no more than $r$. For $\forall \mathbf{x}_{r+1}$ with $\|\mathbf{x}_{r+1}\|_2 = 1$, $(\mathbf{X}^*)^\top \mathbf{x}_{r+1} = \mathbf{0}$, we have $\mathbf{M}^\top \mathbf{x}_{r+1} \neq \mathbf{0}$. Hence, $\mathbf{x}_{r+1}$ is not a valid basis, and we can conclude $\max(\text{rank}(\mathbf{X})) = r$. $\square$

*Proposition* 3 suggests that the target of unknown model fitting can be converted to finding a maximum number of orthometric bases with sparsity constraint. Considering the outliers and noise, the ideal constraints can be written as $\mathbf{M}^\top \mathbf{X} - \mathbf{G} - \mathbf{E} = \mathbf{0}$, where $\mathbf{G}$ and $\mathbf{E}$ are noise and outlier entries, respectively. Following a common perception, $\mathbf{G}$ is assumed to be Gaussian, which suggests to be constricted by *Frobenius* norm $\|\mathbf{G}\|_F$. While outlier matrix $\mathbf{E}$ is sparse in row, and any $\|\mathbf{E}_{i,:}\|_2 \neq 0$ indicates that the $i$-th sample can be considered as an outlier, thus we use $\ell_{2,0}$ to constrain this property. Then our problem is transformed into a multi-objective optimization task, including minimizing the fitting error, the parameter number, the outlier number, and maximizing the basis number. Hence, the general formulation for robust model reasoning and fitting can be written as a unified form:

$$\min_{\mathbf{X}, \mathbf{E}, r} \quad \frac{1}{2}\|\mathbf{G}\|_F^2 + \lambda\|\mathbf{X}\|_0 + \gamma\|\mathbf{E}\|_{2,0} - \tau\text{rank}(\mathbf{X}),$$
$$s.t. \quad \mathbf{M}^\top \mathbf{X} - \mathbf{G} - \mathbf{E} = \mathbf{0}, \quad \mathbf{X}^\top \mathbf{X} = \mathbf{I}_{r \times r}, \tag{14}$$

where $\mathbf{I}$ is an identity matrix of size $r \times r$ with $r = \text{rank}(\mathbf{X})$. It restricts that the bases are all orthometric. $\lambda, \gamma, \tau > 0$ are hyper-parameters to balance different loss items. Problem (14) ultimately outputs an exact sparse subspace $\mathbf{X} \in \mathbb{R}^{D \times r}$, helping to identify the true model $\mathcal{M}$ with $\mathbf{X} \sim \Psi(\mathcal{M})$. In the following, we will explore an efficient solution for this multi-objective problem.

## 2.3 Solution

Problem (14) involves $\ell_0$ minimization, which is the holy grail of sparse approximation. However, $\ell_0$ optimization is NP-hard, and the non-linear objective makes it even more difficult. A practical way is

to use $\ell_1$ norm as the best convex approximation instead. Thus, Problem (14) can be relaxed into:

$$\min_{\mathbf{X}, \mathbf{E}, r} \quad \frac{1}{2}\|\mathbf{M}^\top \mathbf{X} - \mathbf{E}\|_F^2 + \lambda \|\mathbf{X}\|_1 + \gamma\|\mathbf{E}\|_{2,1} - \tau \text{rank}(\mathbf{X}),$$

$$s.t. \ \|\mathbf{x}_i\|_2 = 1, \quad \mathbf{x}_i^\top \mathbf{x}_j = 0, \ \ \forall\, i, j = 1, 2, ..., r, \ \ i \neq j. \tag{15}$$

Another troublesome issue is to maximize $\text{rank}(\mathbf{X})$. Consider that Problem (15) actually aims to obtain $r$ orthometric bases $[\hat{\mathbf{x}}_1, \hat{\mathbf{x}}_2, \cdots, \hat{\mathbf{x}}_r]$ and a sparse outlier matrix $\hat{\mathbf{E}} = [\hat{\mathbf{e}}_1, \hat{\mathbf{e}}_2, \cdots, \hat{\mathbf{e}}_r]$ satisfying $\mathcal{L}(\mathbf{M}, \hat{\mathbf{x}}_i, \hat{\mathbf{e}}_i) = \frac{1}{2}\|\mathbf{M}^\top \hat{\mathbf{x}}_i - \hat{\mathbf{e}}_i\|_2^2 + \lambda\|\hat{\mathbf{x}}_i\|_1 + \gamma\|\hat{\mathbf{e}}_i\|_1 < \tau, \forall i \in \{1, 2, \cdots, r\}$. Thus we decompose (15) into progressively estimating a new sparse bases orthometric to all given bases up to $\mathcal{L}(\mathbf{M}, \hat{\mathbf{x}}_i, \hat{\mathbf{e}}_i) < \tau$ not holds. For any given bases $\{\mathbf{B} = (\mathbf{x}_j)|j = 1, 2, \cdots, i-1\}$ (those have been estimated), a new sparse basis $\mathbf{x}_i(i > 1)$ can be estimated by solving

$$\min_{\mathbf{x}, \mathbf{e}} \mathcal{L}(\mathbf{M}, \mathbf{x}, \mathbf{e}) = \frac{1}{2}\|\mathbf{M}^\top \mathbf{x} - \mathbf{e}\|_2^2 + \lambda\|\mathbf{x}\|_1 + \gamma\|\mathbf{e}\|_1, \quad s.t. \ \|\mathbf{x}\|_2 = 1, \ \mathbf{x}^\top \mathbf{y} = 0, \ \forall \mathbf{y} \in \mathbf{B}. \tag{16}$$

Since there exist two variables $(\mathbf{x}, \mathbf{e})$ to be estimated, it is impractical to optimize them directly with gradient descent method as used in [52, 13, 17]. A valid way is to alternatively optimize $\mathbf{x}$ and $\mathbf{e}$ in an iterative pipeline, which encourages us to compute the closed-form solution for $(\mathbf{x}, \mathbf{e})$ in each iteration. For such purpose, we first calculate the derivatives of objective *w.r.t.* variables, then make them equal to zero. In the $k$-th iteration and for given $\mathbf{x}^{k-1}$, we can easily obtain

$$\mathbf{e}^k = \mathbf{M}^\top \mathbf{x}^{k-1} - \gamma \text{sgn}(\mathbf{e}^k) = \mathcal{T}_\gamma(\mathbf{M}^\top \mathbf{x}^{k-1}), \tag{17}$$

where $\text{sgn}(\cdot)$ is a sign function. $\mathbf{e}^k$ is in fact solved by a standard threshold shrinkage operation $\mathcal{T}_\gamma$ introduced in the *Lemma* of [47]. (Refer to *Append.B* for details)

**Given $\mathbf{e}^k$ Update $\mathbf{x}^k$.** Due to the coexistence of quadratic and sparse constraints, it is hard to directly optimize. Fortunately, [7] provides efficient solution for such linear inverse problem with theoretical and practical verification. We exploit its basic idea into our dual sparsity problem, that is to build at each iteration a regularization of the linearized differentiable function, then obtain

$$\mathbf{x}^k = \arg\min_{\mathbf{x}} \left\{ \frac{L}{2}\|\mathbf{x} - (\mathbf{x}^{k-1} - \frac{1}{L}\nabla f(\mathbf{x}^{k-1}))\|_2^2 + \lambda\|\mathbf{x}\|_1 \right\}, \tag{18}$$

where the smallest Lipschitz constraint of the gradient $\nabla f(\mathbf{x})$ can be calculated by $L(f) = \lambda_{\max}(\mathbf{M}\mathbf{M}^\top)$, and $\lambda_{\max}(\cdot)$ indicates the maximum eigenvalue of a matrix, and we denote

$$\mathbf{q}_L(\mathbf{x}^{k-1}) = \mathbf{x}^{k-1} - \frac{1}{L}\nabla f(\mathbf{x}^{k-1}), \tag{19}$$

where $\nabla f(\mathbf{x}^{k-1}) = \mathbf{M}(\mathbf{M}^\top \mathbf{x}^{k-1} - \mathbf{e}^k)$. In this case, Problem (18) is essentially similar to solving $\mathbf{e}$, and we can obtain the update formula of $\mathbf{x}$ with $\mathbf{x}^k = \mathcal{T}_{\frac{\lambda}{L}}(\mathbf{q}_L(\mathbf{x}^{k-1}))$. (*Proof is in Append.C*)

The above solution for $\mathbf{x}$ actually reduces to a subgradient method [26], which converges at a rate no worse than $\mathcal{O}(1/k)$. However, it was proven in [34, 35] that there exists a gradient method with an $\mathcal{O}(1/k^2)$ complexity which is an 'optimal' first order method for smooth problems [7]. The core strategy is that the iterative shrinkage operation performs at the point $\tilde{\mathbf{x}}^{k-1}$ which uses a specific linear combination of previous two points $(\mathbf{x}^{k-1}, \mathbf{x}^{k-2})$ as $\tilde{\mathbf{x}}^{k-1} = \mathbf{x}^{k-1} + \left(\frac{t_{k-1}-1}{t_k}\right)(\mathbf{x}^{k-1} - \mathbf{x}^{k-2})$ with $t_k = \frac{1}{2}(1 + \sqrt{1 + 4t_{k-1}^2})$. Then updating $\mathbf{x}$ with convergence rate $\mathcal{O}(1/k^2)$ is (See Fig. 2)

$$\mathbf{x}^k = \mathcal{T}_{\frac{\lambda}{L}}(\mathbf{q}_L(\tilde{\mathbf{x}}^{k-1})). \tag{20}$$

Next, we further consider constraints of Problem (16). It can be easily solved by a projection operation [17, 14] in each iteration, including a sphere projection $\mathbf{x} \leftarrow \frac{\mathbf{x}}{\|\mathbf{x}\|_2}$, and an orthogonal projection $\mathbf{x} \leftarrow (\mathbf{I} - \mathbf{B}\mathbf{B}^\top)\mathbf{x}$, because the projector of orthogonal complement is $\mathbf{I} - \mathbf{B}\mathbf{B}^\top$.

Eqs. (17) and (20) provide iterative optimization for searching sparse solution of $\mathbf{x}$ and $\mathbf{e}$, thus we term the whole algorithm as *Dual Sparsity Pursuit* (DSP). Repeatedly using DSP, we can successfully address the general unknown model fitting task, and the whole process is concluded in *Append.D*. Note that, in two-view problem, each model has at least $r = 1$ basis (*e.g.* $\mathbf{F}$) and at most $r = 3$ bases (*e.g.* $\mathbf{H}$), thus we denote $R$ as the max number of bases to be estimated, with $R = 3$.

**Model Reasoning Analysis.** A critical problem is how to reason out the model type and estimate the parameters under a general case. Several methods [44, 41] have been early proposed to use a selection strategy [33, 31, 32], but it is not always clear due their ambiguities [37]. This would be well mitigated in our DSP. *To be specific, for 2D point set fitting, we can directly distinguish the correct model from $\{L, P, E\}$ through the sparsity of estimated $\mathbf{x}$, as shown in Fig. 2. As for two-view models, we can also distinguish them via the sparsity and rank of*

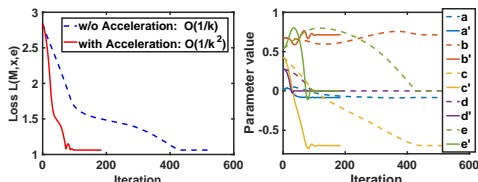

Figure 2: The optimization procedure of our DSP for 2D line fitting. $\mathbf{x} = [a, b, c, d, e]^\top$ and $\mathbf{x}' = [a', b', c', d', e']^\top$ denote the solutions w/o and with acceleration, respectively.

*estimated* $\mathbf{X}$, *referring to the model embedding $\Psi(\mathcal{M})$ or their geometric relations $G_{\mathcal{M}}(D, d, r, s)$.* The following remark may deliver good understanding of this property, which is illustrated in Fig. 4.

**Remark 1** *If the true model is $\mathbf{F}$, the solution for $(\mathbf{x}_2, \mathbf{e}_2)$ would result in a larger fitting error than $(\mathbf{x}_1, \mathbf{e}_1)$, i.e., $\|\mathbf{M}^\top \mathbf{x}_1\|_2 \ll \|\mathbf{M}^\top \mathbf{x}_2\|_2$. In other words, the number of inliers detected by $(\mathbf{x}_2, \mathbf{e}_2)$ would be much smaller than $(\mathbf{x}_1, \mathbf{e}_1)$. Since the second basis is essentially to find a homography or affine structure. On the contrary, if the true model is $\mathbf{H}$ or $\mathbf{A}$, the detected inliers between these two solutions would have small difference, as the orthometric and sparse bases indeed exist.*

## 3 Experiment

**Implementation Details: (1) Parameter setting.** In DSP, $\lambda$ and $\gamma$ are two hyper-parameters. Based on [44], we set $\lambda = 0.005 \log(4N) \times [1, 1, 0.5, 1, 1, 0.5, 0.5, 0.5, 0.1]^\top$ as default (for 2D model, we set $\lambda = \log(2N) \times [0.01, 0.1, 0.1, 1, 1]^\top$). In addition, we set $\gamma = 0.06$ at the beginning, then update it with $0.98\gamma$ for each twenty iterations, and constrain $\gamma_{\min} = 0.02$. Moreover, we set the max iteration as $2k$, and stop it if $\varepsilon = \|\mathbf{x}_k - \mathbf{x}_{k-1}\|_2 \leq 1e{-}6$. As for $\tau$, it controls the number of estimated basis, i.e., $r$. We set $\xi = \mathcal{L}(\mathbf{M}, \mathbf{x}_i, \mathbf{e}_i) / \mathcal{L}(\mathbf{M}, \mathbf{x}_{i-1}, \mathbf{e}_{i-1})$, and describe its distribution on all real data as in Fig. 3. Based on the best $\xi$, we set $\tau = 1.2\mathcal{L}(\mathbf{M}, \mathbf{x}_{i-1}, \mathbf{e}_{i-1})$ during the estimation of $\mathbf{x}_i$. **(2)**

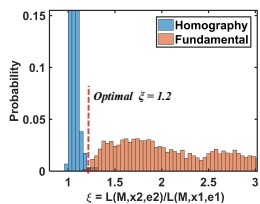

Figure 3: Distribution of $\xi$ on $\mathbf{H}$ data and $\mathbf{F}$ data.

**Synthesized Data.** We synthesize 300 image pairs for each model, which consist of different outlier ratio (OR $= 20\%, 50\%, 80\%$) and noise level (NL $= 0.2, 0.5, 0.8, 1.0, 1.5$ in pixel). **(3) Real Image Data.** 8 public datasets [8, 2] are used, and we divide them into two groups including **Fund**: kusvod2, CPC, TUM, KITTI, T&T; and **Homo**: homogr, EVD, Hpatch. **(4) Comparisons.** 6 methods are used for evaluation, including three SAC methods RNASAC [19], USAC [38], and MAGSAC++ [6]; one globally optimized method EAS [18]; and two deep methods OANet [50], SuperGlue [40]. **(5) Metrics**. All methods applied the normalized 4-point and 8-point algorithms for $\mathbf{H}$ and $\mathbf{F}$ estimation (which are applied as post-process [18] in EAS, DSP and two deep methods with 100 iterations). To measure the accuracy, we use Geometrical Error (GE, in pixel), which is defined as the re-projection error and Sampson distance [4, 5] for $\mathbf{H}$ and $\mathbf{F}$ estimation, respectively. We defined the failed case as that the GE is larger than 5 pixels, or the model is wrongly identified. To avoid the failed cases for a dataset, we use the medium value $E_{med}$ of GE as measurement. (More details are in *Append.E*).

**Model Reasoning and Robust Fitting Test:**
**(1) Test on Synthetic Data.** In the simulation test, an advanced robust estimator EAS [18] is used to estimate all possible models first. Then, three widely used model selection criteria, i.e., AIC [1], BIC [42] and GRIC [25], are equipped. The accuracy (%) of model selection are reported in Tab. 1. AIC can achieve good performance for homography and affine models, but not for epipolar cases, particularly for high OR. BIC shows poor performance in fundamental model selection, since it prefers under-fitting in theory. GRIC fully considers the properties of vision problem performing well in most cases. As for our DSP, it can identify the geometric models with the best accuracy even in case of high outlier ratio.

Table 1: Model reasoning results on syn. data *w.r.t* different OR . Each value $x$ indicates that $x\%$ models are correctly selected.

| OR | Data Model | AIC | BIC | GRIC | DSP (ours) |
|---|---|---|---|---|---|
| 20% | F,100 | 100 | 62 | 100 | 100 |
| | H,100 | 100 | 98 | 100 | 100 |
| | A,100 | 100 | 96 | 100 | 100 |
| 50% | F,100 | 100 | 0 | 100 | 100 |
| | H,100 | 100 | 95 | 100 | 100 |
| | A,100 | 99 | 97 | 98 | 100 |
| 80% | F,100 | 100 | 0 | 93 | 100 |
| | H,100 | 85 | 96 | 98 | 99 |
| | A,100 | 96 | 92 | 95 | 98 |
| all | | 97.2 | 70.7 | 98.4 | 99.7 |

Table 2: Results of model reasoning and fitting on real image pairs. All datasets are divided into Fund. and Homo. based on the model type. Method with * means first estimating all possible models, then using GRIC to select the "best" one. **Bold** and underline indicate the best and second, respectively.

| Data\Methods | | RANSAC* | USAC* | MAGSAC++* | EAS* | OANet* | SuperGlue* | DSP (ours) |
|---|---|---|---|---|---|---|---|---|
| Fund. | $E\_med$ | 0.8478 | 0.6545 | 0.6260 | 0.6900 | 0.7579 | 0.8926 | **0.6017** |
| | Time(s) | 0.5666 | 0.0239 | 0.3918 | 0.2037 | **0.0152** | 0.0721 | 0.0551 |
| | FR | 0.2925 | 0.1992 | 0.2077 | 0.2142 | 0.2632 | 0.2709 | **0.1136** |
| Homo. | $E\_med$ | 1.0428 | 0.8744 | 0.8978 | 0.8472 | 0.8480 | 0.9392 | **0.8227** |
| | Time(s) | 2.010 | 0.0550 | 1.3419 | 0.4463 | **0.0342** | 0.0716 | 0.2794 |
| | FR | 0.3021 | 0.2604 | 0.1424 | 0.0972 | 0.0903 | 0.1181 | **0.066** |
| All | $E\_med$ | 0.8794 | 0.6711 | 0.6383 | 0.7091 | 0.7670 | 0.9026 | **0.6249** |
| | Time(s) | 0.6638 | 0.0260 | 0.4558 | 0.22 | **0.0130** | 0.0721 | 0.0703 |
| | FR | 0.2932 | 0.2043 | 0.2033 | 0.2064 | 0.2515 | 0.2606 | **0.1123** |

**(2) Test on Real Image Data.** In this part, those datasets are divided into Fund and Homo based on their model type. In fact, during real applications, it is more urgent to make sure whether the true model is **F** or **H** [41, 33], thus our used real image data are sufficient to verify the performance for real cases. In addition, considering the competitive performance in Tab. 1, we only use GRIC as comparison in model selection stage. Six representative estimators are used for comparison. That is, first using those robust estimators to obtain all possible models, then using GRIC to select the best one. **Qualitative results** on two representative image pairs are shown in Fig. 4, which obviously reveal the mechanism of our DSP as we assumed in Remark 1. In addition, our algebra results are somewhat coarse, but they can be well refined by a post-processing [18]. **Quantitative results** are evaluated in Tab. 2, including $E_{med}$, *Run Time* (Time) and failure rate (*FR*). From this ta-

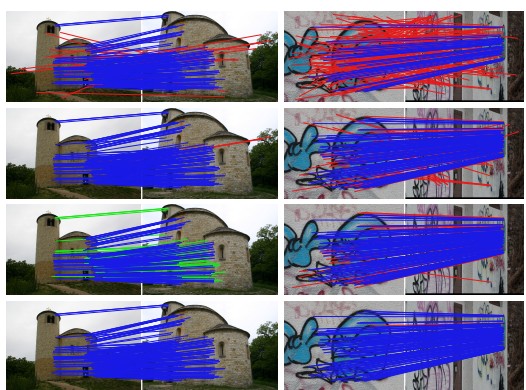

Figure 4: Detected inliers of DSP for real data: **F**(left) and **H**(right). Top to bottom: input data, the first $(\mathbf{x}_1, \mathbf{e}_1)$, second basis $(\mathbf{x}_2, \mathbf{e}_2)$, and refined results. For visibility, at most 200 random selected matches are shown (blue = true positive, green = false negative, red = false positive).

ble, we find that the other estimators typically achieve poor model selection even if using the best criterion GRIC. Three SAC methods get poor model identification, particularly for Homo data, since many image pairs are from extreme view change with high outlier ratios. OANet and SuperGlue similarly achieve low model reasoning accuracy, since some datasets are from complicated scenarios, such as EVD, T&T and CPC, which are not seen by these two deep methods in their training. Their model fitting performances directly affect the model reasoning accuracy. Overall speaking and comparing with GRIC, our DSP reasons out the true model directly from its solutions with much higher accuracy. As for running time, OANet and USAC almost achieve the best efficiency. Because OANet is a deep model with GPU acceleration (SuperGlue is slower since it takes descriptors as input to generate matches), and USAC integrates the local optimization and fast model verification strategy in its universal framework, but sacrifices the accuracy to some extent.

**Known Model Estimation Test:** Known model estimation is the main focus of current researches, thus we also test on those real image pairs by giving the priori of model type, and provide extensive comparisons. The statistic results are shown in Fig. 5, which only contains four challenging datasets, while others are in *Append.E.3*. Statistic results show that the performance of SAC-based methods would heavily degrade with more time consuming and less accuracy on these challenging datasets. This is because they intrinsically have to sample an outlier-free subset to best fit the given model, this is much difficult for these datasets. OANet and SuperGlue achieve poor fitting accuracy due to the same reason particularly for EVD datasets. As for EAS and our DSP, they commonly optimize from global formulation, thus showing similar top accuracy. However, our DSP performs better comparing with EAS, due to the specifical modeling to restrain noise and outliers. And it is much faster, because the use of an acceleration strategy that makes the convergence rate closed to $\mathcal{O}(1/k^2)$.

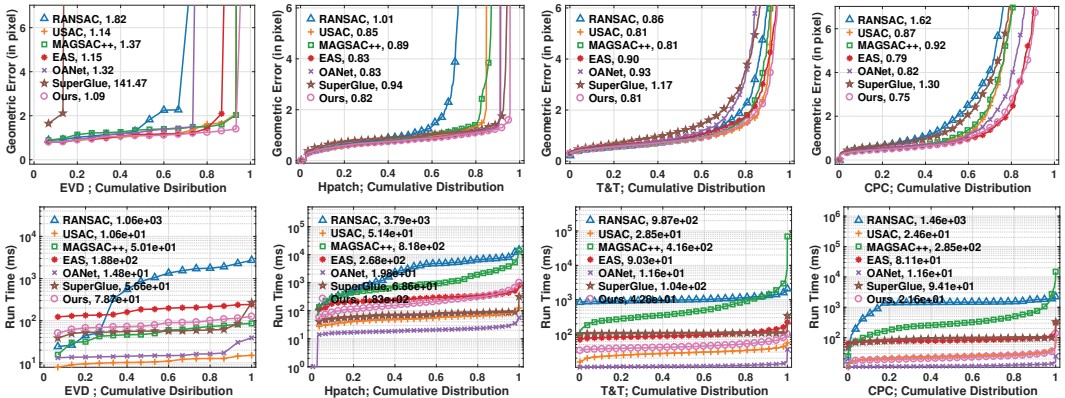

Figure 5: Qualitative statistic results of Geometric Error (top) and Run Time (bottom) for known model estimation. A point on the curve with coordinate $(x, y)$ denotes that there are $100 \times x$ percent of image pairs which have GE or RT no more than y. The lower the better.

**Applications: (1) Multi-model Fitting.** We first consider the multi-class multi-model fitting problem. Unlike multi-model fitting of single type, it would be more challenging if the data contain multiple models with unknown model type and model number. To this end, a public benchmark `AdelaideRFM` is used for evaluation. Quantitative results are reported in Tab. 3, where *Misclassification Error* (ME, %) is selected to characterize the performance. Six algorithms including T-link [29], RCMSA [36], RPA [30], MCT [31], MLink [32] and RFM-SCAN [23], are used for comparison. The table shows that our DSP can achieve the best accuracy with low computational burden.

Table 3: Results of Multi-model fitting. The mean and median values of *ME* (%), and *Running Time* (RT, s) are listed. **Bold** and underline indicate the best and second.

|  | Tlink | RCMSA | RPA | MCT | MLink | RFM-SCAN | Ours |
|---|---|---|---|---|---|---|---|
| *ME_ave* | 27.65 | 10.05 | 5.28 | 11.36 | 6.04 | 2.63 | **1.64** |
| *ME_med* | 27.88 | 6.087 | 4.35 | 1.21 | 4.22 | 1.20 | **0** |
| *RT(s)* | 1.95 | 2.16 | 10.84 | 6.44 | 9.72 | **0.01** | 0.18 |

**(2) Loop Closure Detection.** We next exploit 2 sequences from KITTI vision suite [20] (*i.e.*, K00, K02) to evaluate our method in the loop closure detection (LCD) task, *i.e.*, recognizing re-observations during the navigation of a robot. These two sequences have enough loop closures. Because DELG [10] is considered state-of-the-art for LCD-related tasks such as image retrieval and landmark recognition, we choose it to simultaneously extract global and local features of images to perform LCD in a hierarchical way. Specifically, given a query image, the similarity of global features under L2-metric is first used to select its candidate frame, followed by performing model fitting between them based on local features. Only when sufficient matches are preserved would a loop-closing event be triggered. Results are reported in Tab. 4, which shows that our DSP achieves the best, since our dual sparsity constraints can well address navigation of the degraded cases such as only going ahead or making a turn.

Table 4: Results of LCD. Metric: the recall at 100% precision. **Bold** indicates the best.

|  | RSAC | USAC | MSAC++ | EAS | OANet | Ours |
|---|---|---|---|---|---|---|
| K00 | 0.9112 | 0.9118 | 0.9105 | 0.9086 | 0.7910 | **0.9162** |
| K02 | 0.7632 | 0.7796 | 0.7757 | 0.7664 | 0.6441 | **0.7882** |

## 4  Conclusion

In this paper, we proposed a unified modeling and fast optimization paradigm named DSP for robust model fitting with unknown model type and heavy outliers. It enables to use as few parameters as possible to explain the inlier structure, thus achieving true model identification and parameter estimation. We introduced an alternating optimization strategy together with proximal approximation method to accurately estimate the sparse model parameters and outlier entries. In addition, our proposed method is validated to exactly disambiguate the true model on both synthetic and real data. *Limitation:* Our proposed method can only perform for single model, but real data often contains multiple different models, such as point cloud analysis. In the future, we will integrate the multi-model fitting problem into our modeling, thus achieving a four-fold task in a unified optimization.

## Broader Impact

The proposed method enjoys great potential to improve a wide range of industrial applications including image registration and retrieval, camera pose estimation and localisation, 3D reconstruction, *etc.* To be specific, the proposed method achieves significant accuracy gains of model reasoning and fitting on challenging matching benchmarks, which strongly indicates that it will directly benefit many related fields including robotics and autonomous driving, where the geometric model estimation is the foundation. However, our method does have some unwelcome repercussions like many other machine learning techniques. The accurate and robust model reasoning and estimation can be illegally used for a person or property without permission. It may even be weaponized to guide a UAV to perform self-location then carry out a terrorism attack. But these negative impacts are more related to the fields of application rather than the technology itself. And we believe, under strict supervision, that our work will bring more benefits than harms to society.

## Acknowledgments and Disclosure of Funding

Funding in direct support of this work: NSFC grant 62276192.

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

# Appendices

We provide necessary proof, pseudo code, additional analysis of our method, and more details about the implementation, the used datasets and additional experimental results.

## A    Notaions

We denote matrices with bold upper case letters and their elements with double-indexed lower case letters: $\mathbf{X} = (x_{ij})$. We write the vector in bold lowercase as $\mathbf{x} = (x_i)$. Thus, the matrix can be also termed as $\mathbf{X} = (\mathbf{x}_i)$ or $([\mathbf{X}]_{i,:})$, where $[\mathbf{X}]_{i,:}$ means the $i$-th row of matrix $\mathbf{X}$. We also denote $\mathcal{X}$ as the set-form of matrix $\mathbf{X} \in \mathbb{R}^{M \times N}$, which consists of all column vectors of $\mathbf{X}$, *i.e.*, $\mathcal{X} = \{\mathbf{x}_j\}_{j=1}^N$. The inlier product of two vectors $\mathbf{x}, \mathbf{y} \in \mathbb{R}^n$ is denoted as $\langle \mathbf{x}, \mathbf{y} \rangle = \mathbf{x}^\top \mathbf{y}$. For a matrix $\mathbf{X}$, the maximum eigenvalue is denoted by $\lambda_{\max}(\mathbf{X})$, and its Frobenius norm is defined as $\|\mathbf{X}\|_F = \sqrt{\sum_i \sum_j |x_{ij}|^2}$. For a vector $\mathbf{x}$, $\|\mathbf{x}\|_2$ (or $\|\mathbf{x}\|$) denotes the Euclidean norm $\|\mathbf{x}\|_2 = \sqrt{\sum_i |x_i|^2}$, and its $\ell_0$ norm (*i.e.*, $\|\mathbf{x}\|_0$) is to compute the number of nonzero entries in vector $\mathbf{x}$, while the $\ell_1$ of $\mathbf{x}$ is defined as $\|\mathbf{x}\|_1 = \sum_i |x_i|$.

## B    Threshold Shrinkage Method

Threshold shrinkage method is typically used to solve $\ell_1$ norm problem, which is introduced by the *Lemma* from [47].

**Lemma 1** Let $\mathbf{q}$ be a given vector, for problem:

$$\min_{\mathbf{x}} \frac{1}{2}\|\mathbf{x} - \mathbf{q}\|_2^2 + \lambda \|\mathbf{x}\|_1, \tag{21}$$

if the optimal solution is $\mathbf{x}^*$, then the $i$-th element of $\mathbf{x}^*$ can be calculated as:

$$x_i^* = \mathcal{T}_\lambda(q_i), \tag{22}$$

where $\mathcal{T}_\lambda(\cdot)$ is a threshold shrinkage operation and defined as:

$$\mathcal{T}_\lambda(q) = \begin{cases} q - \lambda, & \text{if } q > \lambda, \\ q + \lambda, & \text{if } q < -\lambda, \\ 0, & \text{otherwise.} \end{cases} \tag{23}$$

## C    Proof of Solution

To solve the following linear inverse problem [7]:

$$\hat{\mathbf{x}} = \arg\min_{\mathbf{x}} \left\{ \frac{1}{2}\|\mathbf{M}^\top \mathbf{x} - \mathbf{e}\|_2^2 + \lambda \|\mathbf{x}\|_1 \right\}, \tag{24}$$

we can extend it into:

$$\min\{F(\mathbf{x}) \equiv f(\mathbf{x}) + g(\mathbf{x})\}, \tag{25}$$

where $f, g$ are convex functions with satisfying the following assumptions:

- $f : \mathbb{R}^n \to \mathbb{R}$ is a smooth convex function, which is continuously differentiable with Lipschitz continuous gradient $L(f)$ :

$$\|\nabla f(\mathbf{x}) - \nabla f(\mathbf{y})\| \leq L(f)\|\mathbf{x} - \mathbf{y}\|_2, \forall \mathbf{x}, \mathbf{y} \in \mathbb{R}^n, \tag{26}$$

  where $L(f)$ is the Lipschitz constant of $\nabla f(\mathbf{x})$.

- $g : \mathbb{R}^n \to \mathbb{R}$ is a continuous convex function which is generally nonsmooth, such as a sparse constraint term for our problem.

- The extended problem (25) is solvable, *i.e.*, $\mathbf{x}^* := \arg\min F \neq \emptyset$.

Table 5: Outline of the embedding of data and model for the classical DLT, and our DSP formulation for known and unknown model fitting. For better understanding, we assume to input only $N$ inliers. For each formulation, the (requirements) including `data type`, `model type`, are listed in the first row. And for each model, we conclude the geometry relationship $G_{\mathcal{M}} = (D, d, r, s)$.

| Model (req.) | DLT (data & model type:D,d,r) | DSP for known model fitting (data & model type:D,d,r) | DSP for unknown model fitting (data type:D) |
|---|---|---|---|
| Fund. | $\mathbf{m}_i^\top = (u_i'\mathbf{p}_i^\top, v_i'\mathbf{p}_i^\top, \mathbf{p}_i^\top)_{1\times 9}$ 
 $\mathbf{x} = vec(\mathbf{F})$ 
 $\mathbf{M}^\top \mathbf{x} = \mathbf{0}_{N\times 1}, G = (9, 8, 1, 0)$ | $\mathbf{m}_i^\top = (u_i'\mathbf{p}_i^\top, v_i'\mathbf{p}_i^\top, \mathbf{p}_i^\top)_{1\times 9}$ 
 $\mathbf{x} = vec(\mathbf{F})$ 
 $\mathbf{M}^\top \mathbf{x} = \mathbf{0}_{N\times 1}, G = (9, 8, 1, 0)$ | $\mathbf{m}_i^\top = (u_i'\mathbf{p}_i^\top, v_i'\mathbf{p}_i^\top, \mathbf{p}_i^\top)_{1\times 9}$ 
 $\mathbf{x} = vec(\mathbf{F})$ 
 $\mathbf{M}^\top \mathbf{x} = \mathbf{0}_{N\times 1}, G = (9, 8, 1, 0)$ |
| Homo. | $\mathbf{m}_i^\top = \begin{bmatrix} \mathbf{p}_i^\top, \mathbf{0}_{1\times 3}, -u_i'\mathbf{p}_i^\top \\ \mathbf{0}_{1\times 3}, \mathbf{p}_i^\top, -v_i'\mathbf{p}_i^\top \end{bmatrix}_{2\times 9}$ 

 $\mathbf{x} = vec(\mathbf{H})$ 

 $\mathbf{M}^\top \mathbf{x} = \mathbf{0}_{2N\times 1}, G = (9, 8, 1, 0)$ | $\mathbf{m}_i^\top = (u_i'\mathbf{p}_i^\top, v_i'\mathbf{p}_i^\top, \mathbf{p}_i^\top)_{1\times 9}$ 

 $\mathbf{X} = \begin{bmatrix} \mathbf{0}_{1\times 3}, \mathbf{h}_3^\top, -\mathbf{h}_2^\top \\ -\mathbf{h}_3^\top, \mathbf{0}_{1\times 3}, \mathbf{h}_1^\top \\ \mathbf{h}_2^\top, -\mathbf{h}_1^\top, \mathbf{0}_{1\times 3} \end{bmatrix}^\top$ 
 $\mathbf{M}^\top \mathbf{X} = \mathbf{0}_{N\times 3}, G = (9, 6, 3, 3)$ | $\mathbf{m}_i^\top = (u_i'\mathbf{p}_i^\top, v_i'\mathbf{p}_i^\top, \mathbf{p}_i^\top)_{1\times 9}$ 

 $\mathbf{X} = \begin{bmatrix} \mathbf{0}_{1\times 3}, \mathbf{h}_3^\top, -\mathbf{h}_2^\top \\ -\mathbf{h}_3^\top, \mathbf{0}_{1\times 3}, \mathbf{h}_1^\top \\ \mathbf{h}_2^\top, -\mathbf{h}_1^\top, \mathbf{0}_{1\times 3} \end{bmatrix}^\top$ 
 $\mathbf{M}^\top \mathbf{X} = \mathbf{0}_{N\times 3}, G = (9, 6, 3, 3)$ |
| Affine | $\mathbf{m}_i^\top = \begin{bmatrix} \mathbf{p}_i^\top, \mathbf{0}_{1\times 3}, -u_i' \\ \mathbf{0}_{1\times 3}, \mathbf{p}_i^\top, -v_i' \end{bmatrix}_{2\times 7}$ 

 $\mathbf{x} = vec(\mathbf{A})$ 

 $\mathbf{M}^\top \mathbf{x} = \mathbf{0}_{2N\times 1}, G = (7, 6, 1, 0)$ | $\mathbf{m}_i^\top = (u_i, v_i, u_i', v_i', 1)_{1\times 5}$ 

 $\mathbf{X} = \begin{bmatrix} a_{11}, a_{12}, -1, 0, a_{13} \\ a_{21}, a_{22}, 0, -1, a_{23} \end{bmatrix}^\top$ 
 $\mathbf{M}^\top \mathbf{X} = \mathbf{0}_{N\times 2}, G = (5, 3, 2, 1)$ | $\mathbf{m}_i^\top = (u_i'\mathbf{p}_i^\top, v_i'\mathbf{p}_i^\top, \mathbf{p}_i^\top)_{1\times 9}$ 

 $\mathbf{X} = \begin{bmatrix} 0, 0, a_{11}, 0, 0, a_{12}, -1, 0, a_{13} \\ 0, 0, a_{21}, 0, 0, a_{22}, 0, -1, a_{23} \end{bmatrix}^\top$ 
 $\mathbf{M}^\top \mathbf{X} = \mathbf{0}_{N\times 2}, G = (9, 7, 2, 5)$ |

In our subproblem, $f(\mathbf{x}) = \frac{1}{2}\|\mathbf{M}^\top \mathbf{x} - \mathbf{e}\|_2^2$, $g(\mathbf{x}) = \lambda \|\mathbf{x}\|_1$. The smallest Lipschitz constraint of the gradient $\nabla f(\mathbf{x})$ can be calculated as

$$L(f) = \lambda_{max}(\mathbf{M}\mathbf{M}^\top). \tag{27}$$

For any $L > 0$, $f(\mathbf{x})$ can be approximated at a given point $\mathbf{y}$ with the following form:

$$\hat{f}(\mathbf{x}) \simeq f(\mathbf{y}) + \langle \nabla f(\mathbf{y}), \mathbf{x} - \mathbf{y} \rangle + \frac{L}{2}\|\mathbf{x} - \mathbf{y}\|^2$$
$$= \frac{L}{2}\|\mathbf{x} - (\mathbf{y} - \frac{1}{L}\nabla f(\mathbf{y}))\|_2^2 + const. \tag{28}$$

By ignoring constant terms related to $\mathbf{y}$, and let $\mathbf{y} = \mathbf{x}^{k-1}$, we can obtain the basic step in each iteration for problem (24):

$$\mathbf{x}^k = \arg\min_{\mathbf{x}} \left\{ \frac{L}{2}\|\mathbf{x} - (\mathbf{x}^{k-1} - \frac{1}{L}\nabla f(\mathbf{x}^{k-1}))\|_2^2 + \lambda\|\mathbf{x}\|_1 \right\}. \tag{29}$$

where $\nabla f(\mathbf{x}^{k-1}) = \mathbf{M}(\mathbf{M}^\top \mathbf{x}^{k-1} - \mathbf{e}^k)$.

# D  Pseudo Code of Our DSP Method

We conclude the pseudo code of the implementation of our DSP method in Alg. 1 and Alg. 2. For any given data, we can directly reason out the true model from the solution of our DSP. To this end, the common embedding and formulation is outlined in Tab. 5. If we know the exact information of true model type, we can also apply our DSP for inlier seeking and model estimation, by giving exact data embedding and basis number $r$, as presented in the middle column of Tab. 5.

Take 2D points fitting as example, *i.e.,* the true model is Line, Parabola, and Ellipse, as shown in Fig. 6. In each case, the inliers are contaminated by noise of 0.01, and the inlier and outlier number are 300 and 100, respectively. We first construct the common data embedding as $\widetilde{\Phi}(x_i, y_i) = \Phi_E(x_i, y_i) = [1, x_i, y_i, x_i^2, y_i^2]^\top$, and the sparse basis or parameter vector is $\mathbf{x} = [a, b, c, d, e]^\top$. We set the max basis number $R = 1$, $\lambda = \log(2N) \times [0.01, 0.1, 0.1, 1, 1]^\top$. The optimization procedure and fitting results are shown in Fig. 6. Specifically, the estimation results for these three cases are $\mathbf{x} = [-0.0845, 0.7135, -0.6956, 0, 0]^\top$, $\mathbf{x} = [-0.4219, -0.1068, -0.4291, 0.7915, 0]^\top$ and $\mathbf{x} = [-0.37825, 0.1498, 0.0641, 0.6513, 0.6373]^\top$, respectively.

---

**Algorithm 1:** Dual Sparsity Pursuit (DSP) Algorithm

---

**Input:** Observation data set $\mathcal{S}$, parameters $\lambda$, $\gamma$, $K$, $\mathbf{B}$, stop threshold $\varepsilon$
**Output:** Sparse basis $\mathbf{x}$ and inlier set $\mathcal{I}$

1  Construct common embedding matrix $\mathbf{M}$ from $\mathcal{S}$ with $\mathbf{m}_i = \widetilde{\Phi}(\mathbf{s}_i)$, and set $\mathbf{e}^0 = \mathbf{0}$, $t_0 = 0$ ;
2  Initialize $\mathbf{x}^0$ as the right singular vector corresponding to the smallest singular value of $\mathbf{M}$ ;
3  Calculate Lipschitz constraint $L = \lambda_{max}(\mathbf{M}\mathbf{M}^\top)$ ;
4  **For** $k = 1, 2, \cdots, K$ **do**
5     Update $\mathbf{e}^k$ using $\mathbf{e}^k = \mathcal{T}_\gamma(\mathbf{M}^\top \mathbf{x}^{k-1})$ and (23) ;
6     Update the gradient $\nabla f(\mathbf{x}^{k-1}) = \mathbf{M}(\mathbf{M}^\top \mathbf{x}^{k-1} - \mathbf{e}^k)$;
7     Update $\mathbf{x}^k$ using $\mathbf{x}^k = \mathcal{T}_{\frac{\lambda}{L}}(\mathbf{q}_L(\tilde{\mathbf{x}}^{k-1}))$ and (23);
8     Perform sphere and orthogonal projection using $\mathbf{x} \leftarrow \frac{\mathbf{x}}{\|\mathbf{x}\|_2}$ and $\mathbf{x} \leftarrow (\mathbf{I} - \mathbf{B}\mathbf{B}^\top)\mathbf{x}$ ;
9     If $\|\mathbf{x}^k - \mathbf{x}^{k-1}\|_2 < \epsilon$, then stop the iteration;
10    Update $t_k$ with $t_k = \frac{1}{2}(1 + \sqrt{1 + 4t_{k-1}^2})$, and compute $\tilde{\mathbf{x}}^k$ as $\tilde{\mathbf{x}}^k = \mathbf{x}^k + \left(\frac{t_k-1}{t_k}\right)(\mathbf{x}^k - \mathbf{x}^{k-1})$ ;
11    Set $\mathbf{x}^k = \tilde{\mathbf{x}}^k$, then perform sphere and orthogonal projection using $\mathbf{x} \leftarrow \frac{\mathbf{x}}{\|\mathbf{x}\|_2}$ and $\mathbf{x} \leftarrow (\mathbf{I} - \mathbf{B}\mathbf{B}^\top)\mathbf{x}$ ;
12 **end for**
13 Return $\mathbf{x} \leftarrow \mathbf{x}^k$, and $\mathbf{e} \leftarrow \mathbf{e}^k$; $\mathcal{I} = \{j | e_j = 0\}$.

---

---

**Algorithm 2:** Model Reasoning & Fitting with DSP

---

**Input:** Observation data $\mathcal{S}$, parameters $\lambda$, $\gamma$, $R$, $K$, stop thresholds $\varepsilon$ and $\xi$
**Output:** Sparse bases $\mathbf{X}$, inlier set $\mathcal{I}$ and model type $\mathcal{M}$

1  Construct common embedding matrix $\mathbf{M}$ from $\mathcal{S}$ with $\mathbf{m}_i = \widetilde{\Phi}(\mathbf{s}_i)$;
2  Initialize $\mathbf{X}_0$ as right singular vectors corresponding to the $R$ smallest singular values of $\mathbf{M}$, $\mathbf{E}_0 = \mathbf{0}$,
   $t_0 = 0$, $\mathbf{B} = \mathbf{0}_{D \times 1}$, $r = 1$ ;
3  Estimate $(\mathbf{x}_1, \mathbf{e}_1)$ using Alg. 1 to solve problem $\min_{\mathbf{x},\mathbf{e}} \mathcal{L}(\mathbf{M}, \mathbf{x}, \mathbf{e})$ ;
4  **For** $i = 2, \cdots R$ **do**
5     Construct given bases $\mathbf{B} \leftarrow [\mathbf{B}, \mathbf{x}_{i-1}]$;
6     Estimate $(\mathbf{x}_i, \mathbf{e}_i)$ using Alg. 1 to solve problem $\min_{\mathbf{x},\mathbf{e}} \mathcal{L}(\mathbf{M}, \mathbf{x}, \mathbf{e})$, and set $\tau = \xi \cdot \mathcal{L}(\mathbf{M}, \mathbf{x}_{i-1}, \mathbf{e}_{i-1})$ ;
7     If $\mathcal{L}(\mathbf{M}, \mathbf{x}_i, \mathbf{e}_i) \geq \tau$, then stop the iteration;
8     Set $r \leftarrow i$ ;
9  **end for**
10 Return $\mathbf{X} = [\mathbf{x}_1, \mathbf{x}_2, \cdots, \mathbf{x}_r]$, $\mathbf{E} = [\mathbf{e}_1, \mathbf{e}_2, \cdots, \mathbf{e}_r]$;
11 Obtain inliers $\mathcal{I} = \{j \mid \|\mathbf{E}_{j,:}\|_2 = 0\}$, and model type $\mathcal{M}$ with $\mathbf{X} \sim \Psi(\mathcal{M})$ .

---

# E  Experiment

## E.1  Implementation Details

We implement the proposed DSP method with MATLAB code. All comparative methods are implemented with the code provided by their authors. Their parameters are set as the original papers suggested, or referring to [6]. The experiments of RANSAC [19], EAS [18] [2] and our DSP are conducted on a desktop with 4.0 GHz Intel Core i7-6700K CPU and 16GB memory. USAC [38] [3], MAGSAC++ [6] [4], OANet [50] [5] and SuperGlue [40] [6] are performed on a Server and Ubuntu 16.04. And two deep learning methods are accelerated by NVIDIA TITAN V GPUs. In addition, AIC [1], BIC [42] and GRIC [25] are used for performing unknown model fitting. Since they are classical model selection criteria, wherein GRIC is still used as a state-of-the-art method in vision tasks [39, 31, 32]. We perform these three criteria with the core parameters suggested as in [31, 32], then equip them with robust model estimators to identify true models.

---

[2]EAS: https://github.com/AoxiangFan/EifficientDeterministicSearch
[3]USAC: https://github.com/cr333/usac-cmake
[4]MAGSAC++: https://github.com/danini/magsac
[5]OANet: https://github.com/zjhthu/OANet
[6]SuperGlue: https://github.com/magicleap/SuperGluePretrainedNetwork

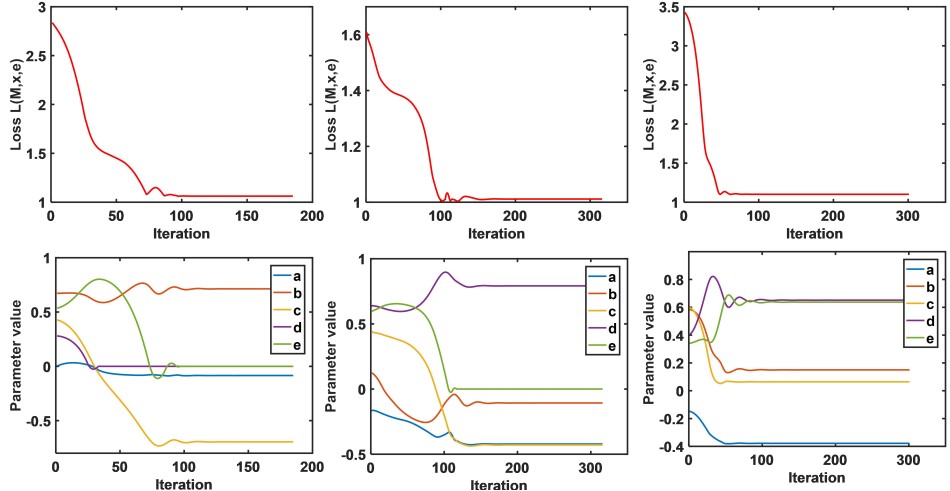

Figure 6: The optimization procedure of our DSP for 2D points fitting. From left to right: model reasoning for a Line, Parabola, and Ellipse model, respectively. From top to bottom: loss $\mathcal{L}(\mathbf{M}, \mathbf{x}_i, \mathbf{e}_i)$ value with iteration, and corresponding parameter values.

### E.2 Details of Datasets

**Synthesized Data.** To perform the robustness test and model reasoning comparisons, we synthesize 900 image pairs with different outlier ratios (*i.e.,* OR $= 20\%, 50\%, 80\%$) and noise levels (*i.e.,* NL $= 0.2, 0.5, 0.8, 1.0, 1.5$ in pixel). To be specific, these synthetic image pairs are randomly generated by two $3 \times 4$ camera matrices $\mathbf{P}_1 = \mathbf{K}_1[\mathbf{I}_{3\times3}|\mathbf{0}]$ and $\mathbf{P}_2 = \mathbf{K}_2[\mathbf{R}_2| - \mathbf{R}_2\mathbf{t}_2]$. Camera $\mathbf{P}_1$ is located in the origin and its image plane is parallel to XY plane. The position of camera $\mathbf{P}_2$ is generated by $\mathbf{R}_2, \mathbf{t}_2$, wherein the position of $\mathbf{P}_2$ is at a random point inside a unit-sphere around $\mathbf{P}_1$, thus $|\mathbf{t}_2| \leq 1$. And the orientation $\mathbf{R}_2 = \mathbf{R}_{X,\alpha}\mathbf{R}_{Y,\beta}\mathbf{R}_{Z,\gamma}$ is determined by three 3D rotation matrices rotating around axes $X, Y$ and $Z$ by $\alpha, \beta$ and $\gamma$, respectively ($\alpha, \beta, \gamma \in [-\pi/3, \pi/3]$). We set $\mathbf{K}_1 = \mathbf{K}_2$ with focal length $f_x = f_y = 600$ and principal points $[300, 300]^\top$. To generate fundamental data, we randomly set $\mathbf{R}_2$ and $\mathbf{t}_2$, and create $n_i$ positions $(x, y, z)$ in real world with $x, y \in [-1, 1], z \in [3, 8]$, which are further projected onto the first and second images by $\mathbf{P}_1, \mathbf{P}_2$. In this case, 20 fundamental image pairs are synthesized for each OR and NL, which creates in total 300 fundamental image pairs. As for generating homography data, we know that homography transformation is typically derived from two types of degraded cases, *i.e.,* the target or scene in shooting is approximated a plane, or the camera is only rotated around optical center without any translation. Therefore and for the first homography case, we randomly simulate $n_i$ positions $(x, y, z)$ in real world with $x, y \in [-1, 1], z \in [9.95, 10.05]$. While for another case, we just set $\mathbf{t}_2 = \mathbf{0}_{3\times1}$ on the basis of synthesizing fundamental data. 10 image data are respectively synthesized for these two homography types for each OR and NL, which also creates in total 300 image pairs. As for generating affine data, we directly set the last row of both $\mathbf{P}_1$ and $\mathbf{P}_2$ as $[0, 0, 0, 1]$, thus making them satisfy the property of affine camera. For each OR and NL, we similarly synthesize 20 affine data, which creates in total 300 affine image pairs finally. For each image pair, we set the number of putative matches as $N = n_i + n_o = 1000$, that means it consists of $n_i = 200$ inliers and $n_o = 800$ outliers if $OR = 80\%$.

**Real Image Data.** 8 public datasets are used for test, including `kusvod2`, `homogr`[7], `EVD`[8], `Hpatch`[9], and the Feature Matching benchmark[10] that contains `TUM`, `KITTI`, `T&T` and `CPC` datasets. The detailed information is concluded in Tab. 6. In particular, `kusvod2`, `homogr` and `EVD` provide putative matches and the inlier indexes. While the rest datasets only provide ground truth model parameters, thus

---

[7]kusvod2, homogr: http://cmp.felk.cvut.cz/data/geometry2view/index.xhtml
[8]EVD: http://cmp.felk.cvut.cz/wbs/
[9]Hpatch: https://github.com/hpatches/hpatches-dataset
[10]FM benchmark: https://jwbian.net/fm-bench

Table 6: Details of the used datasets. From left to right: Datasets name, ground truth(GT) Model, the number of sequences(#Seq), the number of image pairs(#IP), resolution, baseline of the image pair, the property of imaging scenes, and the average match number(AMN) and average inlier ratio(AIR). '–' indicates all image pairs are from different imaging scenes.

| Datasets | GT Model | #Seq. | #IP | Resolution | Baseline | Property | AMN | AIR |
|---|---|---|---|---|---|---|---|---|
| Hpatch | Homo. | 59 | 295 | varying | varying | wall or ground | 3714 | 0.23 |
| EVD | Homo. | – | 15 | $329 \times 278$ to $1712 \times 1712$ | extreme view | wall or ground | 328 | 0.28 |
| homogr | Homo. | – | 16 | $392 \times 278$ to $1712 \times 1368$ | varying | varying | 873 | 0.47 |
| kusvod2 | Fund. | – | 16 | $512 \times 768$ to $1944 \times 2592$ | wide | outdoor scenes | 516 | 0.74 |
| TUM | Fund. | 3 | 1000 | $480 \times 640$ | short | indoor scenes | 333 | 0.62 |
| KITTI | Fund. | 5 | 1000 | $370 \times 1226$ | short | street views | 550 | 0.83 |
| T&T | Fund. | 3 | 1000 | $1080 \times 2048$ *or* $1080 \times 1920$ | wide | outdoor scenes | 892 | 0.35 |
| CPC | Fund. | 1 | 1000 | varying | short | internet photos | 491 | 0.35 |

we use VLFEAT toolbox[11] to construct SIFT matches with ratio test 1.5, and the inlier matches are identified as the geometric error less than 2 pixels.

### E.3 Additional Results

**Quantitative Results of Other Four Datasets.** We first give additional results of the known model fitting on the rest four datasets, including homogr, kusvod2, KITTI and TUM. Details are in Fig. 7, which show the same trends as in other four datasets (please see the main body), that our proposed DSP achieves the best fitting accuracy and promising efficiency, followed by EAS and SAC methods.

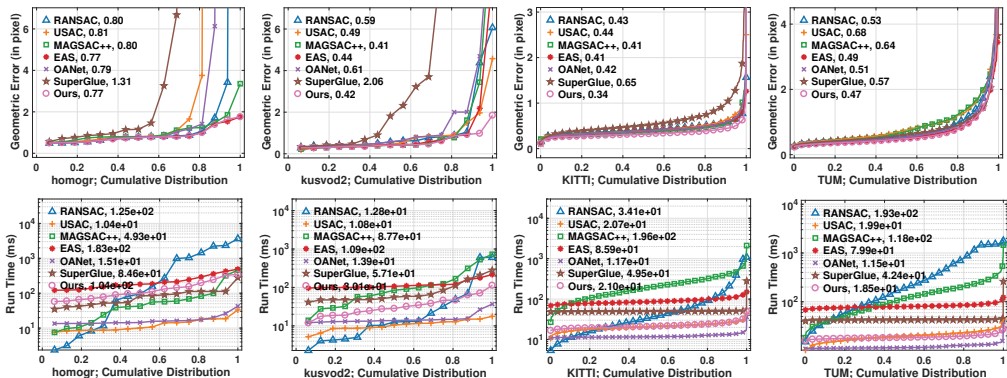

Figure 7: Qualitative statistic results of Geometric Error(top) and Run Time (bottom) for known model estimation. A point on the curve with coordinate $(x, y)$ denotes that there are $100 \times x$ percent of image pairs which have Error or Time no more than y. The lower the better.

### E.4 Applications

**Multi-class Multi-model Fitting.** In this part, we apply our DSP to solving multi-class multimodel fitting problem. In multi-model fitting of single type, the multiple structures belong to the same model type, such as being only line or circle, thus making it convenient for predefining the model type then conducting fitting algorithm. But this would be troublesome if the data contains multiple geometrical models of different types, such as simultaneously existing line, parabola, and circle, or both fundamental and homography model in two-view geometry. Existing strategy commonly borrows selection criterion like GRIC, which usually requires a long runtime [31, 32]. Next, we tend to demonstrate the superiority of our DSP in terms of simultaneously model reasoning and fitting in the task of multimodel fitting problem.

We first test DSP on synthesized point data, which consists of three inlier structures including line, parabola and ellipse together with a large number of outliers. The target is to classify each inlier structure and identify those outliers. Several manifold subspace clustering and multi-model fitting

---

[11]VLFeat: https://www.vlfeat.org/

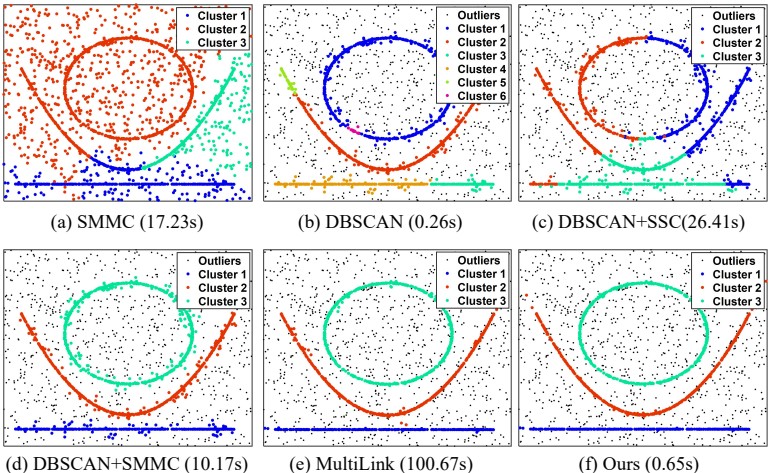

(a) SMMC (17.23s)  (b) DBSCAN (0.26s)  (c) DBSCAN+SSC(26.41s)

(d) DBSCAN+SMMC (10.17s)  (e) MultiLink (100.67s)  (f) Ours (0.65s)

Figure 8: Multi structure recovering and inlier seeking on synthetic data, which consists of three inlier structures (including ellipse, parabola and line with noise level 0.05, respectively generated by $500, 500$ and $300$ inliers) and randomly generated outliers of number $1000$, that means the total number of points is 2300. SMMC [48], SSC [15], DBSCAN [16], MultiLink [32] and their combinations are used for comparison. The runtime in second is attached in each sub-figure.

methods are used for comparison, such as spectral clustering on multiple manifolds (SMMC) [48], sparse subspace clustering (SSC) [15], density-based clustering (DBSCAN) [16], and a recently published preference-based method termed MultiLink [32] that uses GRIC for model selection.

Clustering results are shown in Fig. 8, we can find that our proposed DSP achieves the best accuracy by combining with DBSCAN. Since our DSP is merely designed for unknown single model fitting, which encourages us to integrate multi-model problem into our outlier rejection, model selection and fitting framework and achieve joint optimization in the future. Multi-Link can well classify these inlier structures as well, but several outliers that are closed to inliers are wrongly classified due to its fixed thresholds. Since MultiLink asks for repeat estimations for all possible models, and uses agglomerative clustering as final output, it may consume a huge runtime (100 seconds for 2300 points). In addition, DBSCAN can identify several potential inlier structures and filter out the most outliers with a short runtime. But the performance of DBSCAN is much sensitive to its core parameters. In other words, a loose parameter would

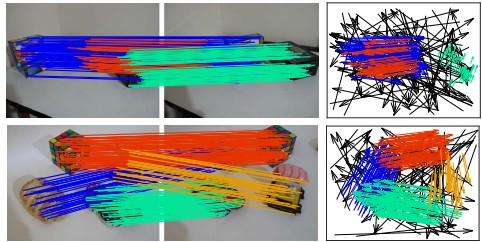

Figure 9: Multi-model fitting results on real images of our DSP method. In each scene, different colors denote different inlier structures, and the head and tail of each arrow in the motion field correspond to the positions of feature points in two images. For visibility, at most 200 randomly selected matches are presented, and the true negatives are not shown. Best view in color.

easily classify neighboring outliers as inlier structure, while a strict setting prefers to classify one complete structure into several small clusters, as shown in subfigure (b). In theory, SMMC and SSC can not handle outliers, and failed in clustering the data of a high number of outliers, which can be seen in subfigure(a). In this regard, we combine the advantages of DBSCAN and some clustering methods. We find that SMMC can well refine the results of DBSCAN to be more accurate, but SSC can only work for liner structures such as line or plane, thus fails in our synthesized data.

We also extend our method on real multi-model fitting data, *e.g.* AdelaideRFM [12]. Fig. 9 shows some qualitative results with three structures and four structures, which demonstrates that our DSP can accurately cluster the corrupted data into several inlier structures together with an outlier cluster. Quantitative results are reported in the main body manuscript.

---

[12]https://cs.adelaide.edu.au/~hwong/doku.php?id=data

