# OpenReview forum: "Robust Model Reasoning and Fitting via Dual Sparsity Pursuit"
_NeurIPS.cc/2023/Conference — NeurIPS 2023 spotlight_

### Official Review · Reviewer_bMPj · 2023-07-02

**Soundness:** 4 excellent
**Presentation:** 4 excellent
**Contribution:** 4 excellent
**Rating:** 8
**Confidence:** 4

**Summary:**

This paper is about model fitting. The authors consider a scenario in which it is unknown whether correspondences between points in 2 images stem from 3D points that are (1) generally distributed, (2) lie on a plane, or (3) lie on a plane and the motion between the 2 images is not projective but affine. The latter implies that not a pinhole camera was used for projection but an affine camera. This can hold if, for instance, the distance between camera and scene is large compared to the depth variation within the scene.

Case (1) amounts to the well-known fundamental matrix constraint x’^T*F*x=0 with (x, x’) being corresponding 2D homogeneous points. Here, F is a 3x3 matrix with rank(F)=2. For noise-free data, matrix F can be estimated from at least 8 point correspondences since F is determined only up to scale and each point-point correspondence yields a single equation.
Case (2) amounts to a 3x3 homography H. Differently from the F-matrix, homographies have full-rank, in general. As fundamental matrices, they are only determined up to scale. Since each point-point correspondence yields 2 equations, H can be determined from at least 4 correspondences.
For a general affine transformation between the two images, case (3) implies that the last row of H equals [0,0,1]. There are no constraints on the remaining 6 entries, in general.

The authors show that by expressing cases (2) and (3) as fitting multiple subspaces, all three cases can be expressed by the same model. They propose a projected coordinate descent type of algorithm to estimate.


**Strengths:**

To me, this interpretation of F/H-matrix geometry is new. I cannot say whether this naturally arises from classical theorems. The fact that all papers I am aware of fit either an F-matrix or a homography can be taken as evidence that at least this fact is not generally known.

**Weaknesses:**

- proposition 2 is wrong
Matrix Psi has rank 2 which can be seen from the fact that the third row is linearity dependent on the first two. Hence, G_H=(9,7,2,3).



**Questions:**

Please correct proposition 2.

---

> ### Author Rebuttal · Authors · 2023-08-09
>
> # Q: About the embedding of homography matrix and its geometric relationship.
>
> **R:** We would like to thank the reviewer’s positive feedback.
>
> It seems that the reviewer was concerning about proposition 1. As for proposition 1, we agree that the homography constraint derives two independent equations, since the last one is a linear combination of the first two. But this happens for DLT solution, that converts to fitting a hyperplane $\theta = vec(\mathbf{H})$. Under our subspace recovery framework, it converts to estimate subspace $\Psi(\mathbf{H})$, which is full rank of 3, thus the third basis indeed exists, and we have $G_{\mathbf{H}} = (9,6,3,3)$. This is also verified and applied in related papers [A][B] (such as Eq. (2) or Tab. 1 in [A]).
>
> _[A] Robust Homography Estimation via Dual Principal Component Pursuit. CVPR 2020._
>
> _[B] Efficient Deterministic Search With Robust Loss Functions for Geometric Model Fitting. TPAMI 2022._

---

> > ### Comment · Reviewer_bMPj · 2023-08-18
> >
> > I thank the authors for their explanation. I do not have any further questions.

---

### Official Review · Reviewer_WgUj · 2023-07-06

**Soundness:** 2 fair
**Presentation:** 2 fair
**Contribution:** 3 good
**Rating:** 6
**Confidence:** 2

**Summary:**

The paper studies the geometric model fitting problem with unknown model type and heavy outliers. It proposes a unified optimization model with dual sparsity constraints that combines the outlier rejection, true model reasoning and parameter selection. Moreover, a fast numerical algorithm is proposed to solve the approximate and dimension-reduced model via separability of the objective function. Numerical experiments on synthetic and real data sets are conducted to compare the performance of the proposed method with other related works. Overall, the paper is complete from theory, algorithm, to experiments with a section of broader impact.

**Strengths:**

1. The proposed dual-sparsity optimization model has a certain novelty.
2. A variety of numerical results are presented to justify the proposed effectiveness.

**Weaknesses:**

1. The existence and uniqueness of the proposed model are not discussed in detail.
2. The description of formulation for the proposed model is not fully concise and precise, especially the treatment of the rank-term.
3. Model sensitivity and robustness could be further discussed from the theoretical aspect.

**Questions:**

1. Even with the Proposition 3, model (13) is not equivalent to model (14) which is later solved inexactly. Why was the convex relaxation  applied to (14) rather than (13)? Would that cause loss of accuracy especially the low rankness?
2. How about the uniqueness of the solutions for the proposed model?
3. Convergence analysis for the proposed algorithm could be provided.

**Limitations:**

The proposed work aims to avoid parameter selection by introducing a variable of parameters in the unified model. However, it leads to the introduction of other regularization parameters associated with the sparsity terms in the objective function, which again may need fine tuning. Some practical guidance for diverse data sets and application settings should be highlighted.

---

> ### Author Rebuttal · Authors · 2023-08-09
>
> # Q1: About problem (13), (14), and the rank-term?
> **R1:** For better understanding, we  should introduce  problem (13) first, which is actually modeled for finding all $r$ independent sparse bases, by supposing $r$ is known. But for our unknown model fitting task, $r$ is unknown in advance, thus needs to be estimated simultaneously.  For such purpose, we introduced Proposition 3, which converts the estimation of $r$ into $r = \max rank(\mathbf{X})$. Hence we integrated this into our general formulation, i.e., obtaining problem (14). This conversion is natural and necessary. And both the later convex relaxation and solution are based on problem (14), because (14) is our finally general formulation, while (13) is merely a middle formulation.
>
> # Q2. How about the uniqueness of the solutions for the proposed model?
> **R2:** We will discuss the global optimality of our proposed DSP, that implies the existence and uniqueness of solution.
>
> Our DSP formulation actually follows the minimization problem on Grassmannian $\mathbb{G}(r,D)$. An element of $\mathbb{G}(r,D)$ can be represented by an orthonormal matrix (multiple bases) in $\mathbb{O}(r,D) = : \{ X \in \mathbb{R}^{D\times r}: X^T X = I_r\}$, which is the well-known Stiefel manifold. With this understanding, we can obtain the following parameterized problem:
>
> $\min\limits_{X \in \mathbb{O}(r,D)} f(X)$,
>
> where $f:\mathbb{R}^{D\times r}\rightarrow \mathbb{R}$
> is locally Lipschitz, possibly non-convex and non-smooth. In our DSP formulation, we have
>
> $f(X) = \frac{1}{2}\|M^T X -E\|_F^2 + \lambda ||X||_1 + \gamma ||E||_2$$_1$.
>
> Since  we consider problems on the Grassmannian, we use tools from Riemannian geometry to state optimality conditions. Specifically, we generalize the definition of the Clarke subdifferential and denote $\tilde{\partial}f$ the Riemannian subdifferential [D] of $f$:
>
> $\tilde{\partial}f(X)= (I-XX^T)\nabla f(X)$.
>
> We say that $X$ is a critical point of our DSP problem if and only if $0=\tilde{\partial}f(X)$, which is a necessary condition for being a global optima. Note that $f$ is non-smooth but convex, and the Clarke subdifferential is a nonempty and convex set since a locally Lipschitz function is differentiable almost everywhere. Thus we can conclude the global optima $X^*$ exists.
>
> Actually, in the community of robust subspace recovery, DPCP and its follow-ups [A]-[C]  have provided theoretical guarantees for modeling, the property of global optima and convergency. These researches can also provide basic theory for our DSP formulation, directly giving theoretic guidance for our own global optimality analysis.
>
> # Q3: Convergence analysis for the proposed algorithm could be provided.
> **R3:**  Intuitively, because our objective is convex and we use a gradient-descent-like optimization method, it must converge into the optima under proper initialization. Specifically, given a constant step size, suppose that function $f$ satisfies the $(\alpha,\epsilon,X^*)$-Riemannian Regularity Condition. Let $\\{X_k\\}$ be generated by our DSP with step size $\mu_k \equiv \mu \leq \alpha\epsilon/\\xi^2 $ and initial $X_0$ satisfying $dist(X_0, X^*)\leq \epsilon$, where $\epsilon$ denotes an upper bound on the size of the Riemannian subgradients in a neighbrohood of $X^*$, and $\alpha>0$ denotes a scale factor.
> Then, for all $k>0$, we have
> $dist(X_k,X^*)\leq max\\{ dist(X_0,X^*) - \mu \alpha k/2,\mu\xi^2/\alpha \\} $,
> which implies that after at most $K=2(dist(X_0,X^*)-\mu\xi^2/ \alpha)/(\mu\alpha)$ iterates, the inequality $dist(X_k, X^*)\leq \mu\xi^2/ \alpha$ will hold for all $k\geq K$, i.e.,converge on $K$. A larger $\mu$ leads to a faster decrease but a larger universal upper bound of $\mu\xi^2/ \alpha$. Papers [A]-[D] could provide more details of convergence analysis for such problem.
>
> # Q4: About Parameter settings.
> **R4:** The hyper parameters of our DSP do not need fine tuning for each scene, since our original purpose is to explore a unified method for all unknown models under different scenes. These parameters have exact physical significance, which provides a guidance for parameter settings.
> * In details, $\gamma$ is used to constrain outliers, indicating that error entry $\mathbf{e}$ generates if two vectors are not vertical to some extent. In other words, a correct match should satisfy
>  $|cos(\theta(\mathbf{m}_i, \mathbf{x}))| <= \gamma$,
> where $\theta(\mathbf{m}_i, \mathbf{x}) $ denotes the angles of two vectors. For correct matches without noise, we have $\mathbf{m}_i^T\mathbf{x} = 0$, i.e., they are vertical.
>  Under noise case, we set $\gamma = 0.05$, which allows an inlier have $\theta(\mathbf{m}_i, \mathbf{x}) \in [88.85^{\circ},91.15^{\circ}]$.
>
> We also follow the idea of simulated annealing, during  optimizing, we dynamically decrease $\gamma = 0.98\gamma$ for each 20 iterations and set $\gamma_{min} = 0.02$.
>   * As for parameter $\lambda$, it balances the fitting error and model complexity. Prof. Philip HS Torr has studied this in paper ''Geometric motion segmentation and model selection'', and concluded that under the assumption of independence among matches, the optimal estimation is merely contributed by 4 noisy coordinate values of each match $(u_i , v_i , u_i' , v_i ')$, thus the parameter item and error item would differ by a scale factor of $log(4N)$.
> * Threshold $\tau$ is used to control the estimation of basis number $r$, Fortunately, the optima value of $\tau$ can derive from the statistical analysis result as shown in Fig. 3.
>
> _[A] Dual Principal Component Pursuit:Improved Analysis and Efficient Algorithms, NeurIPS 2018._
>
> _[B] A Linearly Convergent Method for Non-Smooth Non-Convex Optimization on the Grassmannian with Applications to Robust Subspace and Dictionary Learning, NeurIPS 2019._
>
> _[C] Dual Principal Component Pursuit for Robust Subspace Learning:Theory and Algorithms for a Holistic Approach, ICML 2021._
>
> _[D] Subgradient Descent Learns Orthogonal Dictionaries, ICLR 2019._

---

> ### Comment · Area_Chair_MyNm · 2023-08-21
>
> Dear Reviewer WgUj,
>
> Thank you for being a reviewer for NeurIPS2023, your service is invaluable to the community!
>
> The authors have already submitted their feedback and I noticed that you don't appear to have submitted a new round of comments.
>
> Could you examine rebuttals and other reviewers' comments, and open up discussions with the authors and other reviewers?
>
> Regards, Your AC

---

> ### Comment · Reviewer_WgUj · 2023-08-21
>
> All my comments have been addressed and no more further questions pop up at this point, so I will raise the score.

---

### Official Review · Reviewer_cPqq · 2023-07-06

**Soundness:** 3 good
**Presentation:** 3 good
**Contribution:** 2 fair
**Rating:** 6
**Confidence:** 4

**Summary:**

This paper considers the robust model fitting problem in the presence of outliers, which is a fundamental problem in low-level CV. The aim is to simultaneously achieve outlier rejection, model selection, and model parameter estimation in a unified formulation. Toward this end, the authors propose to cast the joint outlier rejection, model selection, and model estimation problem into a sparse subspace recovery problem, which can cover the widely used projective transformation models for multi-view gemometry such as the fundamental, homography and affine models. The joint optimization formulation is solved by an alternating algorithm with the use of proximal approximation computation and acceleration. Experimental results on synthetic and real-word data have been provided to deminstrate the performance of the new method, including fundamental matrix and homography estimation, as well as a loop closure application.

**Strengths:**

This work is well motivated to joitnly achieve outlier rejection, model selection, and model estimation in a unified formulation. The proposed sparse subspace recovery formulation covers the widely used projective transformation models for multi-view gemometry, such as the fundamental, homography and affine models. The method has been evaluated on both synthetic and real-world data via various experiments.

**Weaknesses:**

1. The formulation only applies to algebraic error model, i.e., model estimation with algebraic distance. While algebraic distance is convenient due to its linearity, geometric distance is geometrically or statistically meaningful and can yiled better performance in projective transformation estimation over the algebraic distance. This has been demonstrated in multi-view geomety.

2. Wile the proposed method is well conceived, it would not outperform the simple method that first estimates the model parameters of each candidate models and then selects the best model in terms of the fitting error. Although the provided experiments show that it outperforms AIC, BIC, GRIC selection methods, in practical applications the model is typically selected in terms of a score of the estimated model computed based on the symmetric transfer errors when the ground-truth transformation is unknown. This selection method is typically used in practice.

3. The proposed algorithm has several hyper-parameters, e.g., $\gamma$, $\lambda$, $r$, and $\tau$. Its performance depends on the tunning of these hyper-parameters, which diminishes its potential advantage over the simple model selection method based on the symmetric transfer errors.

**Questions:**

It is claimed that the objective with geometric error is extremely hard to optimize due to the highly non-linear nature, but in fact there exist well developed methods in the literature of multi-view geomety that can solve the formualtion with geometric error efficiently and effectively.

**Limitations:**

The authors have not addressed the limitations.

---

> ### Author Rebuttal · Authors · 2023-08-09
>
> # Q1: About using geometric distance.
> **R1:** We highly agree that minimizing geometric error (GE) could obtain better performance in accuracy, since the error entry with geometric distance is more stable. But the non-convex nature indeed makes it hard to optimize. Although there exist effective methods to search the optima, it usually consumes complex computation and long execution time, thus rarely used in real applications, such as SLAM, 3D Reconstruction, etc.
>
> To our knowledge, GE is commonly converted into a tractable form, such as a generalized fractional form (GFF) or Sampson Distance (SD, i.e., first order approximation). For instance, Ref. [A] studies approximate solutions for GE-based objective using GFF for each match $(p,p’)$ like:
> $GE(p,p’) = \frac{f(p,p’)} {g(p,p’)}$, where $f$, $g$ are linear functions.
> In Sec. 6.2.1 of [A], the authors also admitted that _''finding least squares estimates based on geometric distances is intractable''._
> Actually they merely explored GE minimization for Homography and Affine matrix, as the authors explained in Sec 6.1.2 _''Fundamental matrix estimation does not have the GFF'',_ hence they still used Algebraic Error (AE) to estimate $F$ matrix.
>
> As for SD, we take fundamental matrix $F$ as example:
> $GE(F, p, p’) = \frac {(p’^TFp)}  {(F^Tp’)^2_1 + (F^Tp’)^2_2 + (Fp)^2_1+ (Fp)^2_2} $.
> Obviously, using SD to estimate $F$ is not easy, at least more difficult than using AE. In fact, under consensus maximization framework, such as RANSAC, GE is typically used to count inliers with pixel threshold. But the model parameters are still estimated by AE. Recently, SD often serves as loss to guide the training of those deep matching methods.
>
> In addition, by reading the experimental results from Tab. 3 and Tab. 4 of [A], we clearly find, GE-based optimization would take $>$5000ms to process around 500 point matches with 50% inlier ratio. While our DSP costs only 183ms in average for Hpatch datasets (average match number 3714 and average inlier ratio 23%), and merely causes 0.82 pixels Geometric Error. Paper [B] (Fig. 6) also reveals that, GE-based method (IBCO) [C] costs huge runtime.
>
> Another critical point is that, the core motivation of our manuscript is to explore a unified modeling and efficient solution for unknown model fitting. But, the GE forms would vary for each specific model, thus hard to be integrated into our unified objective to achieve model reasoning.
>
> Considering both the practicability and theory convenience, it is better to use Algebraic Error in our DSP. Surely, if the reviewer has any efficient and effective methods/literature to solve GE-based problems, please recommend them to us. It would help to advance our dual sparsity formulation in the future, i.e., modeling with GE and achieving better accuracy while maintaining the real-time property.
>
> _[A] Deterministic approximate methods for maximum consensus robust fitting. TPAMI 2021._
>
> _[B] Efficient Deterministic Search With Robust Loss Functions for Geometric Model Fitting. TPAMI 2022._
>
> _[C] Deterministic consensus maximization with biconvex programming, ECCV 2018._
>
>
> # Q2: Details of using GRIC.
> **R2:** In our experiments, we did use geometric error to support AIC, BIC and GRIC to conduct model selection. In details, for all comparing methods that use AIC, BIC, or GRIC for model identification, we first utilize those robust estimators (including RANSAC, USAC, MAGSAC++, EAS, OANet, SuperGlue) to estimate each model $\mathcal{M}$, then use $\mathcal{M}$ back to compute the geometric error for each match pair. In our experiments, we use Sampson Distance as geometric error with Matlab toolbox.
>
> # Q3: About the difficulty of tuning hyper parameters $\gamma,  \lambda, r$ and $\tau$.
> **R3:** The hyper parameters of our DSP do not need fine tuning for each scene, since our original purpose is to explore a unified method for all unknown models under different scenes. These parameters have exact physical significance, which provides a guidance for parameter settings.
>
> * In details, $\gamma$ is used to constrain outliers, indicating that error entry $\mathbf{e}$ generates if two vectors are not vertical to some extent. In other words, a correct match should satisfy
>  $|cos(\theta(\mathbf{m}_i, \mathbf{x}))| <= \gamma$,
>  where $\theta(\mathbf{m}_i, \mathbf{x}) $ denotes the angles of two vectors. For correct matches without noise, we have $\mathbf{m}_i^T\mathbf{x} = 0$, i.e., they are vertical.
> Under noise case, we set $\gamma = 0.05$,
>  which allows an inlier have
>  $\theta(\mathbf{m}_i, \mathbf{x}) \in [88.85^{\circ},91.15^{\circ}]$.
>
> We also follow the idea of simulated annealing, during  optimizing, we dynamically decrease $\gamma = 0.98\gamma$ for each 20 iterations and set $\gamma_{min} = 0.02$.
>
>  * As for parameter $\lambda$, it balances the fitting error and model complexity. Prof. Philip HS Torr has studied this in paper “Geometric motion segmentation and model selection”, and concluded that under the assumption of independence among matches, the optimal estimation is merely contributed by 4 noisy coordinate values of each match $(u_i , v_i , u_i' , v_i ')$, thus the parameter item and error item would differ by a scale factor of $log(4N)$.
>
> * $r$ is not a hyper parameter, but a core integer variable that we need to estimate. It indicates the maximum number of basis of recovered subspace or geometric model. In our DSP, we constrain it with rank maximization and solve it progressively.
>
> * The estimated basis number $r$ is controlled by threshold $\tau$, but fortunately, the optima value of $\tau$ can derive from the statistical analysis result as shown in Fig. 3.

---

> > ### Comment · Reviewer_cPqq · 2023-08-20
> >
> > The authors have addressed most of my previous concerns, so I raise my rating to 6.

---

### Official Review · Reviewer_qrzv · 2023-07-06

**Soundness:** 3 good
**Presentation:** 3 good
**Contribution:** 4 excellent
**Rating:** 8
**Confidence:** 5

**Summary:**

Considering that existing model estimation methods highly rely on the correct definition of model types, this paper introduces a unified optimization modeling DSP to simultaneously reason out the model type and estimate model parameters from contaminated data. For such purpose, the authors proposed Sparse Subspace Recovering (SSR) theory to modeling geometric model estimation task, that is to search a maximum of independent sparse bases under an over-embedded data space. The authors also introduced a fast and robust solver to estimate the sparse subspace parameters and error entries, and validated the advanced performance of their method on both unknown and known model estimation, and two applications.

**Strengths:**

- This paper is well written and organized, and easy to follow.
- The motivation, theory, formulation and solution are good contributions for the model fitting topic.
- The authors solved the geometric model estimation problem from a novel perspective, that considers the model reasoning task additionally. Particularly, the authors introduced Sparse Subspace Recovering theory and formulated the unknown model fitting task into a continuous optimization objective, and explored efficient solution for it.
- The authors designed unknown model fitting experiments, and also evaluated their method on common exact model fitting task, which is reasonable and credible. The experiments are convincing, and the results show great superiority comparing to the SOTA.


**Weaknesses:**

- Line 74, the authors claimed that if the data are properly normalized, using algebraic error is good. But how to assure this property of input data?
- This paper proposes SSR theory, but the authors have not given mathematical explanation or proof.
- As for the solution, why not using ADMM to solve this problem, I think ADMM is a common choice for solving L1 norm problem with Lagrange multiplier, as sparse subspace clustering (SSC) used. Please explain.
- Line 52: How to understand ``insufficient information’’ that the GRIC used, which may cause wrong model selection for constrained motions. What are the advantages of this method comparing with those model selection criteria.
- If using DSP or SSR theory to estimate a Fundamental matrix F, how to use the rank 2 constraint of F? That is an intrinsic property of Fundamental matrix.

Typos:
- Line 60, add citation for PSGM when it first appears.
- Lines 163 and 171, ``sparse independent hyperplanes``, I think it is better to use ``sparse intersected hyperplanes`` or ``sparse independent bases``.
- Line 170, $\mathcal{R}$ should be $\mathbb{R}$ to indicate the real number space.
- Line 226, $e_k$ --> $e^k$.
- Line 239, ``can successfully addressing`` --> can successfully address.
- Line 271, ``300 image pairs of each model`` --> ``300 image pairs for each model``.



**Questions:**

- Explain how to ensure the input data properly normalized.
- Please provide mathematical explanation or proof for your SSR theory.
- Please explain or validate the necessary of the used solution, or compare it with ADMM.
- Please emphasize the advantages of your DSP comparing with existing model selection strategies, such as GRIC.
- Correct the typos, see [Weaknesses]


**Limitations:**

Yes, the authors have analyzed the limitations and potential negative social impact.

---

> ### Author Rebuttal · Authors · 2023-08-09
>
> # Q1: How to ensure the input data properly normalized?
> **R1:** The related distribution of the input data is their intrinsic nature, we cannot change it.  But to ensure the estimation easier, we first normalized the input points of each image into zero mean and one standard error, then we scaled each point into length 1 using $\mathbf{m}_i \leftarrow \mathbf{m}_i/||\mathbf{m}_i||_2 $  .
>
>
> # Q2: Mathematical explanation of Sparse Subspace Recovery (SSR) theory.
> **R2:** Our SSR theory claims that ‘’Geometric model fitting can be seen as a subspace recovery problem represented by the intersection of multiple sparse hyperplanes under an over embedded data space.’’,  which can be interpreted by the following mathematical form:
>
> Suppose we are given a set of clean data $\mathcal{S} =\\{ \mathbf s_i\\}_{i=1}^N$,
> sampled from a geometric model $\mathcal{M}$,
>
> which can be parameterized by several bases $\\{\theta_j\\}_{j=1}^r$.
> On this basis, we can obtain compact constraint $f_j(\mathbf{s}_i)^{T}\theta_j = 0,~j = 1,2,\cdots,r$,
> where $f_j(\mathbf{s}_i)$ is a specific embedding for point $\mathbf{s}_i$.
>
> Our SSR theory suggests that, it can be extended into $\{[f_j(\mathbf{s}_i)^{T}, g_j(\mathbf{s}_i)^{T}][\theta_j^T,\mathbf{0}^T]^T=0,~j=1,2,\cdots,r\}$. Here $g_j(\mathbf{s}_i)$ is another data embedding that is not contained in $f_j(\mathbf{s}_i)$. $\mathbf{0}$ is an all zero vector with length equal to $g_j(\mathbf{s}_i)$. For example, a line model derives $f(\mathbf{s}_i)= [x,y,1]^T, \theta = [a,b,c]^T$, with $\{f(\mathbf{s}_i)^{T}\theta = 0 \}$ under ideal case, note that here $r= 1$, thus we ignore subscript $j$. Then we generate  $g(\mathbf{s}_i) = [xy, x^2,y^2]^T$, and get an over embedded space $F(\mathbf{s}_i)=[f_j(\mathbf{s}_i)^{T}, g_j(\mathbf{s}_i)^{T}] = [x,y,1,xy, x^2,y^2]$, and $\theta' = [\theta,0,0,0]^T$, similarly having $F(\mathbf{s}_i)^{T}\theta ' =0$.
>
> Based on the argument embedding $g_j(\mathbf{s}_i)$, we can construct a common embedding $F(\mathbf{s}_i)^{T}$ for multiple bases, such that $\{F(\mathbf{s}_i)^{T}\theta_j' = 0, j =1,2,\cdots, r\}$, then obtain our general formulation Eq. (9) for unknown model fitting, i.e., $ \widetilde{\Phi}(\mathbf{s}_i)^T\Psi(\mathcal{M})= \mathbf{0}$.
>
> SSR theory can be proven easily. For each basis, since $f_j(\mathbf{s}_i)^{T}\theta_j = 0$ holds, $g_j(\mathbf{s}_i)^{T}\mathbf{0} = 0$ equally holds, thus we have
> $\{[f_j(\mathbf{s}_i)^{T}, g_j(\mathbf{s}_i)^{T}][\theta_j^T,\mathbf{0}^T]^T = 0,~j = 1,2,\cdots,r\}$.
>
>
> # Q3: Why not using ADMM to solve this problem?
> **R3:** Using ADMM to solve our DSP problem requires complex computation, including the inversion operation for large matrix, which creates long execution time. Specifically, ADMM suggests an auxiliary variable $Z$ to simplify nonlinear constraint, and we have:
>
> $\min\limits_{(X,E,Z)} \frac{1}{2} ||M^TX-E||_F^2 $
> $+\gamma ||E||_2$$_1 +\lambda ||Z||_1$
>
> $ s.t.~X^TZ = I ,X = Z.$
>
> Then, the Augmented Lagrangian expression is:
>
> $L= \\frac{1}{2}  ||M^TX-E||_F^2$
> $+\gamma  \\|E\\|_2$$_1+ \lambda \\|Z\\|_1$
> $+ <\Lambda_1,X^TZ - I> + <\Lambda_2, X-Z> + \frac{\mu_1}{2} ||X^TZ - I||_F^2 +  \frac{\mu_2}{2} ||X-Z||_F^2,$
>
> $=\frac{1}{2}||M^TX-E||_F^2 +\gamma ||E||_2$$_1+ \lambda \\|Z\\|_1$
> $+\frac{\mu_1}{2} ||X^TZ - I + \frac{\Lambda_1}{\mu_1} ||_F^2 + \frac{\mu_2}{2} ||X-Z+\frac{\Lambda_2}{\mu_2}||_F^2 + const.$
>
> Letting $\frac{\partial L}{\partial X} = 0$,
>  we have the update formula for $X^{k+1}$  at given $(E^k,Z^k,\Lambda_1^k,\Lambda_2^k，\mu_1^k,\mu_2^k))$,
>
> $ X^{k+1} = [MM^T + (\mu_1+\mu_2) I]^{-1} [ME^{k} + \mu_1^k (I-\frac{\Lambda_1^k}{\mu_1^k})Z^T + \mu_2^k(Z - \frac{\Lambda_2^k}{\mu_2^k} )] $
> which is time consuming. In addition, parameters $(E^k,Z^k,\Lambda_1^k,\Lambda_2^k，\mu_1^k,\mu_2^k))$ also need update.
>
>
> # Q4: Our DSP V.S. GRIC.
>
> **R4:** GRIC is an early strategy to tackle unknown model fitting problem, but still widely used in many vision tasks. Given the model pool $\mathcal{M}$, traditional pipeline first uses existing robust estimators, such as RANSAC, to estimate the parameters for each model in $\mathcal{M}$, then uses GRIC to select the “best” one as final output. Using this estimation-then-selection strategy, the time consumption would be much large if the model pool is huge. And this greedy strategy would easily cause wrong identification, as revealed in Tab. 1. On the contrary, our proposed DSP fully considers the forms under sparse subspace, and specifically models the noise and outliers, thus solving three subproblems in a unified optimizing paradigm, such that our DSP achieves better accuracy and efficiency.
>
>
> # Q5: How to ensure rank 2 constraint of F model.
> **R5:** During our optimization, we do not consider the rank 2 constraint for $F$ at the beginning,  that would obtain a full rank matrix $F'$. Then we decompose $F' $ with SVD, getting $F'= U diag(s1,s2,s3)V^T$, the final estimation of rank-2 form would be $\hat{F} = U diag(s1,s2,0)V^T$.  This process is commonly done in many robust estimators such as 8-point algorithm.
>
> # Q6: Typos.
> **R6:** We will correct them in our next version.

---

> ### Comment · Reviewer_qrzv · 2023-08-19
>
> All of my concerns have been thoroughly addressed in the rebuttal. In general, the authors have presented a novel perspective in addressing the problem of geometric model estimation. Notably, their contributions lie in the effective modeling of unknown model fitting and the efficient exploration of solutions. The writing and experimental part are also commendable. Therefore, I wholeheartedly recommend accepting this manuscript.

---

### Official Review · Reviewer_1N35 · 2023-07-11

**Soundness:** 4 excellent
**Presentation:** 4 excellent
**Contribution:** 4 excellent
**Rating:** 7
**Confidence:** 5

**Summary:**

This paper addresses the task of robust model reasoning and fitting in an unified optimization framework that can estimate the geometric model accurately without knowing the predefined model in advance while being robust to outliers as well as highly efficient. The authors propose a novel sparse subspace recovery theory and derive corresponding propositions, thus giving a general and unified formulation for robust model reasoning and fitting. They next introduced an alternating optimization strategy together with proximal approximation method to accurately estimate the sparse model parameters and outlier entries. Extensive experiments indicate that the proposed method outperforms the selected comparison methods.

**Strengths:**

This paper solved the geometric model estimation problem from a novel perspective, that is recovering the model parameters without knowing the model type, which is interesting and valuable. The writing is good. The authors have provided clear presentation to convey their core idea. The experiment part is reliable, which well verifies the advantages of the method.

**Weaknesses:**

1:The authors designed an accelerated optimization approach and demonstrated its fast convergence in Fig. 2, claiming it an ‘optimal’ first order method. But there are no experimental proofs of the optimality in the paper. Authors should compare the performance of the algorithm before and after the acceleration.

2:The authors mentioned their strength in efficiency, but the experiment results have not revealed this property.

3:The authors propose a solution process similar to DPCP, and I think it should be added to the comparison algorithms as well.

4:The experiments in the paper demonstrate the excellent performance of the DSP on 2D models and geometry models. I would like to know is the upper bound of the algorithm. Is it possible to design experiments (e.g., continuously increasing the dimension of the data space) to test the performance upper bound.

5:During application test, the proposed method have successfully applied to multimodel fitting, as we can see. But why do the authors mention that they plan to integrate multimodel fitting and achieve a four-fold task in the future?

6:The pose estimation or visual localization experiments are also necessary, since they are more common in this topic, comparing with multimodel fitting or Loop Closure Detection.

7:Others:
 Line 113, “to predefine” should be “predefining”
 Line 159, “convert” should be “be converted”


**Questions:**

Please refer to [Weaknesses]

**Limitations:**

The authors have solved most limitations of their method, and they also mentioned potential negative social impact, which are acceptable.

---

> ### Author Rebuttal · Authors · 2023-08-09
>
> # Q1: About the optimality if using the Acceleration Strategy (AS).
> **R1:** The optimality has not changed if using AS. We can see from Fig. 2 that, at convergence stage, solution or loss value are identical for using AS or not. This is because our used AS is not an approximation for original problem that may discard partial accuracy. Instead, it borrows the concept of Momentum or Nesterov method in gradient decent, to use an adjusted startpoint at each update step. It was proven in [7][30][31] that this strategy can achieve $O(1/k^2)$ convergence rate with theoretic guarantees, that is an ‘‘optimal’’ first order method for smooth problem.
>
> # Q2: Emphasize the efficiency.
> **R2:** We will explain the strength of efficiency as follows. First, EAS [TPAMI 2022] and our DSP are both global optimizing paradigms, but our DSP just consumes half time than EAS, due to the use of AS. Second, SAmpling Consensus (SAC) methods require sufficient time budget to hit an all-inlier subset, which is highly related to outlier ratio and noise scale. Thus, our DSP is faster for the cases of high outlier ratio, and this trend is well revealed in Fig. 3 of Supp. Mat. As for USAC, it integrates the local optimization and fast model verification in a universal framework, thus achieving fast model estimation, but most of the results are coarse or incorrect. Third, two deep methods just perform one forward propagation once trained, and they are accelerated with GPUs, thus obtain the best efficiency.
> **Without GPUs, the average execution time of OANet increases from 13ms to 117.6ms (our DSP: 70.3ms)**. In addition, compared with above methods, our DSP can estimate geometric model from contaminated data without predefining correct model type.
>
> # Q3: Comparing with DPCP
> **R3:** Our solution is merely partially similar to DPCP. DPCP provides plain formulation and solution for robust subspace learning with theoretic guarantees. But applying it to geometric model needs additional consideration, such as in DPCP-H [CVPR2020] and EAS [TPAMI2022]. In particular, EAS explores more general formulation and efficient solution for each model type, thus we take it as a representative method of DPCP and select it for comparison. As for the optimization process, because our DSP additionally considers the constraints of noise and the complexity of subspace, our formulation is more complex and hard to optimize. Thus, we explore an alternating optimization framework with proximal approximation strategy to accurately estimate the sparse subspace and outlier entries. To tackle the constraint terms, we use the similar strategy as DPCP, that performs sphere projection and orthogonal projection for orthogonal constraint $X^TX = I$.
>
> # Q4: Applying to high-dimensional model.
> **R4:** Our method mainly focuses on geometric models, particularly two-view geometry, to reason out model type and estimate model parameters from corrupted data. Thus, applying to recover high-dimensional or more complex models still need further exploration.
>
> # Q5: About multimodel fitting.
> **R5:** Our DSP have not realized multi-model fitting in a unified formulation yet, instead it asks for density-based clustering method to obtain several coarse clusters first, then apply our DSP for each cluster to reason out accurate model type and parameters, thus largely enhancing the fitting performance. In this regard, we think it is valuable to integrate estimating model number into our DSP formulation to achieve a four-fold task in the future.
>
> # Q6: Evaluation for pose estimation or visual localization.
> **R6:** In our experiment, we have conducted Fundamental matrix and Homography matrix estimation, which can directly reveal the performance of these two mentioned applications. Because the camera pose is directly decomposed from estimated $F$ matrix, i.e., with known camera intrincics $K_1, K_2$, the relation between $F$ and camera pose $R, t$ is $[t]_{\times}R = E = K_2^{-T}FK_1^{-1}$.
>
>  Following the reviewer's suggestion, we also conduct pose estimation on large scale dataset YFCC100M. This dataset has ground truth poses and sparse models obtained from an  off-the-shelf SfM tool. We follow the setting in OANet (Zhang et al., 2019) and choose 4 sequences for evaluation. We detect up to 2k matches for each data with SIFT. Similar to OANet, we use AUC of the pose error at threshold $(5^{\circ}, 10^{\circ},20^{\circ})$ for evaluation, where the pose error is defined as the maximum of the angular errors in rotation and translation. Our DSP can obtain the best performance among those handcrafted methods.
>
> | Method    |  $@5^{\circ}$  | $@10^{\circ}$  |  $@20^{\circ}$  |
> | ---- |  ----  | ----  | ----  |
> | RANSAC| 3.47  |9.10 |  18.60 |
> | USAC | 5.67  | 9.53  | 13.41 |
> | MAGSAC++ | 11.80  | 19.72 | 33.15|
> | EAS            | 12.18  | 22.22 | 35.67 |
> | DSP(Ours) | 14.43  |  25.32|   37.73|

---

> ### Comment · Reviewer_1N35 · 2023-08-19
>
> In the rebuttal, the authors have effectively addressed my concerns regarding their proposed acceleration strategy, the design of comparative methods, and the applications. These responses have provided me with a deeper understanding of the method. And, I believe that the proposed theory of unknown model fitting and its corresponding solution will make good contributions to the field of computer vision. Therefore, I keep my rating as 7.

---

### Decision · Program_Chairs · 2023-09-21

**Decision:**

Accept (spotlight)

**Comment:**

This work has been reviewed by five reviewers and received consistent positive scores (SA/WA/SA/WA/A), with four reviewers unequivocally acknowledging the novelty of the proposed dual-sparsity optimization model and excellent performance in the experimental evaluations. Overall, the AC considers that the paper makes a solid contribution to the field and recommends acceptance of the paper.